# Statistical analysis supports pervasive RNA subcellular localization and alternative 3' UTR regulation

Rob Bierman[1], Jui M Dave[2], Daniel M Greif[2], Julia Salzman[1,3]*

[1]Department of Biochemistry Stanford University, Stanford, United States; [2]Department of Biomedical Data Science Stanford University, New Haven, United States; [3]Departments of Medicine (Cardiology) and Genetics Yale University, New Haven, United States

## eLife assessment

This paper describes an **important**, well-organized study into an under-exploited area of spatial transcriptomics. The limitations of the approach are generally made clear, but there is insufficient orthogonal validation to demonstrate the biological significance of the results, which leads to the evidence for the claims being currently **incomplete**. Nevertheless, the tools presented will provide a resource to researchers wishing to characterise spatial patterning of mRNAs, and the paper will be of interest to researchers studying cell biology, RNA biology, and method development for spatial transcriptomics/proteomics.

*For correspondence:
julia.salzman@stanford.edu

Competing interest: The authors declare that no competing interests exist.

**Abstract** Targeted low-throughput studies have previously identified subcellular RNA localization as necessary for cellular functions including polarization, and translocation. Furthermore, these studies link localization to RNA isoform expression, especially 3' Untranslated Region (UTR) regulation. The recent introduction of genome-wide spatial transcriptomics techniques enables the potential to test if subcellular localization is regulated in situ pervasively. In order to do this, robust statistical measures of subcellular localization and alternative poly-adenylation (APA) at single-cell resolution are needed. Developing a new statistical framework called SPRAWL, we detect extensive cell-type specific subcellular RNA localization regulation in the mouse brain and to a lesser extent mouse liver. We integrated SPRAWL with a new approach to measure cell-type specific regulation of alternative 3' UTR processing and detected examples of significant correlations between 3' UTR length and subcellular localization. Included examples, *Timp3*, *Slc32a1*, *Cxcl14*, and *Nxph1* have subcellular localization in the mouse brain highly correlated with regulated 3' UTR processing that includes the use of unannotated, but highly conserved, 3' ends. Together, SPRAWL provides a statistical framework to integrate multi-omic single-cell resolved measurements of gene-isoform pairs to prioritize an otherwise impossibly large list of candidate functional 3' UTRs for functional prediction and study. In these studies of data from mice, SPRAWL predicts that 3' UTR regulation of subcellular localization may be more pervasive than currently known.

## Introduction

As a general rule, it is accepted that the cellular localization of a protein is biologically critical for its function (*Hung and Link, 2011*). However, the general importance of RNA localization within a cell, and how this localization varies in different biological situations remains poorly understood. Targeted studies have identified examples of genes whose RNA localization is critical to function, such

as the enrichment of beta-actin (*Actb*) RNA to lamellipodia in motile chicken embryonic myoblasts (*Lawrence and Singer, 1986*). It was observed that approximately 80% of total actin mRNA localized to the lamellipodia, and specific disruption of localization, but not expression, of the mRNA, resulted in decreased cell motility (*Kislauskis et al., 1994*; *Kislauskis et al., 1997*). The same authors also identified so-called 'zipcode' sequences in the 3' UTR of *Actb* which were necessary for proper RNA localization (*Kislauskis et al., 1994*). In a larger-scale study, it has been estimated that 70% of mRNAs are spatially localized in *Drosophila* embryogenesis (*Lécuyer et al., 2007*). Other well-known and recently identified examples of RNA subcellular localization with functional consequences include lipid droplets (*Saka and Valdivia, 2012*) and *TIS11B* protein granules (*Ma and Mayr, 2018*). In these case studies, RNA localization is cis-regulated by either alternative splicing or 3' UTR usage.

While the vast majority of 3' UTR isoform functions remain unknown and incompletely annotated, emerging evidence points to an abundance of cell-type specific regulation (*Meyer et al., 2022*) where the inclusion of different 3' UTRs may even have opposite functions. *Cd47*, for example, expresses a long-isoform 3' UTR that results in a peripherally localized protein product protecting against phagocytosis, but can also express a short-isoform 3' UTR that results in a cytoplasmic protein product with the same amino-acid sequence that does not confer the same phagocytotic protection (*Berkovits and Mayr, 2015*). Control of RNA subcellular localization through RNA isoform choice may help pinpoint functions for alternative RNA isoforms and UTRs in eukaryotes.

Spatial transcriptomics has seen rapidly increasing interest as methods become increasingly powerful and affordable (*Marx, 2021*). However, work remains primarily focused on gene expression. Techniques such as MERFISH (*Moffitt et al., 2016*), and its commercialization Vizgen, as well as SeqFISH+ (*Eng et al., 2019*) utilize sequential multiplexed fluorescence imaging to localize hundreds to thousands of distinct genes across a tissue with subcellular resolution. Along with RNA-capture-based spatial transcriptomics techniques (*Ståhl et al., 2016*; *Stickels et al., 2021*; *Su et al., 2021*), these spatial datasets have primarily been used to analyze the distribution of cell-types within a tissue via gene expression. At a finer scale, RNA distribution within cells has been understudied despite an established history of biologically important case studies discussed in multiple reviews (*Lipshitz and Smibert, 2000*; *Holt and Bullock, 2009*; *Suter, 2018*).

The limited approaches that have been used to detect subcellular localization patterns from high throughput, high-resolution spatial datasets rely on co-stains and/or heuristics without statistical formalism (*Samacoits et al., 2018*; *Xue et al., 2020*; *Tang et al., 2021*). As an example, an analysis of a SeqFISH + dataset relied on arbitrarily chosen hard thresholds to determine peripherally and centrally localizing genes in different mouse cortex cell-types. The use of thresholding can result in overlooked weaker spatial patterns and also makes it difficult to control the false discovery rate (FDR) (*Eng et al., 2019*). Additionally, compartment-based analysis of MERFISH datasets has been used to detect differences in neuron soma, axon, and dendrite transcriptomes using the Wilcoxon rank-sum test and Moran's I (*Moran, 1950*; *Xia et al., 2019*). Discretizing cellular regions does not fully utilize the information present in the MERFISH dataset since RNA subcellular localization is intrinsically a continuous process. Similarly, while proximity-tagging and sequencing approaches such as APEX-seq (*Fazal et al., 2019*; *Padrón and Ingolia, 2022*) have generated high-plex datasets for RNA localization within subcellular compartments, these methods require genetically modified cell-lines, and cannot be readily applied to tissue. Finally, to our knowledge, no study has attempted to test whether isoform regulation can explain subcellular localization at the gene level in massively multiplexed FISH datasets.

To address the limitations of prior approaches, we introduce Subcellular Patterning Ranked Analysis With Labels (SPRAWL) as a transparent and statistical approach to detect RNA subcellular patterning from multiplexed imaging datasets. SPRAWL assigns an interpretable score to detect RNA localization patterning for a gene of interest in an individual cell. Furthermore, these scores can be carefully aggregated to detect spatial patterns between cell-types and biological replicates with FDR control. SPRAWL currently identifies continuous peripheral, central, radial, and punctate localization patterns which are significantly more extreme than expected by chance in either direction of effect. SPRAWL can be extended to detect user-defined patterns and represents a general framework for unbiased discovery of RNA subcellular localization patterns from multiplexed imaging datasets. This integrative approach identifies genes with potential cis-regulatory spatial sequences, and prioritizes candidates for experimental follow-ups.

# Results

SPRAWL was developed to be a non-parametric single-cell resolved measure of RNA subcellular localization that is robust against confounding variables of cell size, and RNA expression level, while providing effect-size and statistical significance measures. SPRAWL reduces complex spatial patterns into one-dimensional scores that are readily interpretable and comparable. An additional benefit of SPRAWL scores is their direct integration with other statistical methods: scores can be analyzed through the lens of various metadata such as cell type, or correlated with other measures such as RNA 3′ UTR regulation or splicing state.

SPRAWL is a publicly available Python package that can be installed using pypi with pip install subcellular-sprawl and has also been implemented in Nextflow (*Di Tommaso et al., 2017*) and Docker for reproducible analyses at large scale in high-performance or cloud computing environments. SPRAWL source code and documentation are available on GitHub, (copy archived at *Bierman, 2024*).

## SPRAWL quantifies peripheral and central subcellular RNA patterning with rank statistics

Examples of RNA localized to the plasma membrane include *Actin* and *Tubulin* in mammalian cells (*Lawrence and Singer, 1986*), *ASH1* in yeast (*Bertrand et al., 1998*), and *Oskar* in fly oocytes (*Rongo*

a) Rank all RNA spots by boundary

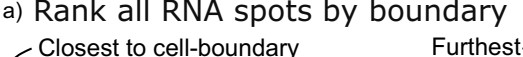

r = median rank of gene of interest

• RNA molecule from gene of interest
· RNA molecule from other genes

$n$ = Total RNA spots
$g$ = Unique genes
$m$ = RNAs per gene

$$\sum_{i \in g} m_i = n$$

b) PMF of median rank R under $H_0$

$$P(R \leq r) = \sum_{k=1}^{r} \frac{\binom{k-1}{\frac{m-1}{2}}\binom{n-k}{\frac{m-1}{2}}}{\binom{n}{m}}$$

c) SPRAWL peripheral score

$$X = \frac{R - \frac{n+1}{2}}{\frac{n+1}{2} - 1}$$

Limits and Behavior
$$-1 \leq X \leq 1 \qquad E[X] = 0$$

SPRAWL score interpretation

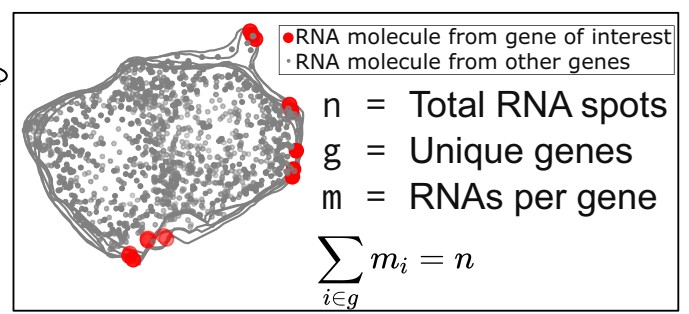

Peripheral    Random    Anti-peripheral

$x = 0.94$    $x = 0.02$    $x = -0.90$

d) Cell-type peripheral significance

$$Y = \left(\sum_{k=1}^{n} \sigma_k^2\right)^{-\frac{1}{2}} \sum_{k=1}^{n}(X_k - \mu_k) \xrightarrow{d} N(0,1)$$

Endothelial cells with gene of interest marked in red

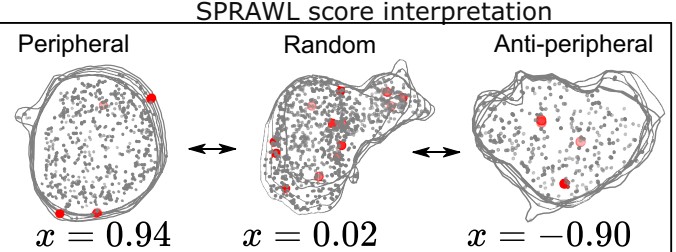

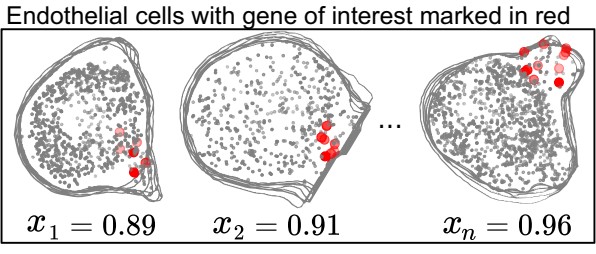

$x_1 = 0.89$    $x_2 = 0.91$    $x_n = 0.96$

**Figure 1.** Subcellular Patterning Ranked Analysis With Labels (SPRAWL) peripheral and central score workflow. (**a**) RNAs are ranked from closest to furthest from the cell-boundary to calculate the median peripheral rank of the gene of interest. For the central metric, distances from the cell centroid are used for ranking instead. (**b**) Under the null hypothesis of each rank being equally likely, the probability mass function of the median is exactly calculable. (**c**) The intuitive SPRAWL score per gene per cell, X, will be near +1 for highly-peripheral patterns, near 0 for randomly-peripheral patterns, and near –1 for anti-peripheral patterns. (**d**) Peripheral significance of a gene within a cell-type is estimated from per cell SPRAWL scores using the Lyapunov Central Limit Theorem (CLT). Overlaying cell outlines are a result of viewing 3D slices from the top down.

The online version of this article includes the following figure supplement(s) for figure 1:

**Figure supplement 1.** Subcellular Patterning Ranked Analysis With Labels (SPRAWL) metrics have high specificity and lack bias.

*et al., 1995*). These foundational examples motivate the unbiased statistical detection of RNA localization patterns in reference to the cell-boundary. To satisfy this need, we've created the SPRAWL peripheral metric (*Figure 1*) which quantifies the extent to which the RNA spots of a gene of interest are more extremely proximal or distal from the cell-membrane than expected by chance.

To calculate the SPRAWL peripheral metric for a given gene in a given cell, first, the minimum euclidean distance is calculated between each RNA spot, regardless of gene identity, and the cell-boundary. These distances are then used to rank the spots from 1 to n corresponding to the nearest and furthest RNA spot from the boundary, respectively (*Figure 1a*). The median rank is calculated for the m RNA spots of the gene. Under the null hypothesis that the gene is not peripherally localized, the expected value is (n+1)/2. Genes with lower median ranks than the expected value are more peripherally localizing, while larger median ranks correspond with anti-peripheral localization.

The probability mass function (PMF) of observing each possible median peripheral rank has a direct formulation which allows for exact calculations of p-values under the null (*Figure 1b*). The actual SPRAWL peripheral score, X, is the result of normalizing the median rank to be between –1 (anti-peripheral) and 1 (peripheral) with an expected value of 0 (not peripheral) (*Figure 1c*). Finally, the per cell-type scores can be calculated as the mean of the SPRAWL cell scores to provide an aggregate measure, Y, of RNA localization per gene per cell-type. Under the Lyapunov Central Limit Theorem (*Billingsley, 1995*), Y will approach in distribution a standard normal as the number of cells increases. The SPRAWL centrality score is conceptually identical to the peripheral score, but RNA spots are ranked by distance from the cell-centroid rather than the cell boundary. All subsequent steps are the same as the peripheral metric.

One of the main advantages of using a rank-based formulation of the periphery and centrality scores is the insensitivity to cell size and rotation. This feature facilitates direct comparisons of SPRAWL scores between cells and even samples. The simplicity of the statistically-backed metrics provides both effect size and p-value handles for detecting extreme RNA patterning in either the positive (peripheral/central) or negative (anti-peripheral/anti-central) direction of effect. Finally, it is worth noting that while the peripheral and central scores are strongly anti-correlated (*Figure 1—figure supplement 1c*), there are clear examples of RNA with simultaneously central and peripheral localization in a cell when the cell-boundary runs near to the cell centroid.

## SPRAWL detection of punctate and radial patterning relies on gene-label permutations

While some RNAs are known to be peripherally or centrally localizing as discussed above, others are known to be trafficked to organelles (*Chang et al., 2004*), cell-poles (*Rongo et al., 1995*; *Hachet and Ephrussi, 2004*), or neuronal processes (*Minis et al., 2014*; *Zappulo et al., 2017*; *Das et al., 2019*). In all cases, RNA molecules of the same gene will be more spatially aggregated than expected by chance. To detect such patterning, SPRAWL punctate and radial metrics have been defined to respectively identify RNA species that tend to aggregate by euclidean distance or in one angular sector of the cell.

SPRAWL's punctate score represents the degree to which RNA spots from a given gene are clustered together, scores closer to 1 indicate self-colocalizing or self-aggregating genes. Scores near –1 indicate self-repulsion, and scores of 0 indicate an expected level of aggregation under the null of random patterning.

When calculating the punctate score for a gene of interest with m>1 RNA spots in a cell, a subset of k random pairs of spots are selected and the distances between them are measured and averaged (*Figure 2a*). Next gene-label permutations are performed, randomly swapping gene labels but not RNA spot locations, to create a null background of mean between-spot distances by again choosing k random spots from the gene of interest in each permuted cell (*Figure 2b*). The punctate score, X, is normalized to be between –1 and 1 with E[X]=0 under the null (*Figure 2c*). Negative values indicate anti-punctate patterning, values near 0 are random or non-punctate, and positive values indicate punctate behavior (*Figure 2d*). Finally, SPRAWL cell-type scores can be calculated using the Lyapunov Central Limit theorem in the same manner as in the peripheral score (*Figure 1d*). The radial metric is conceptually identical to the punctate metric but measures mean between-spot angles instead of between-spot distances.

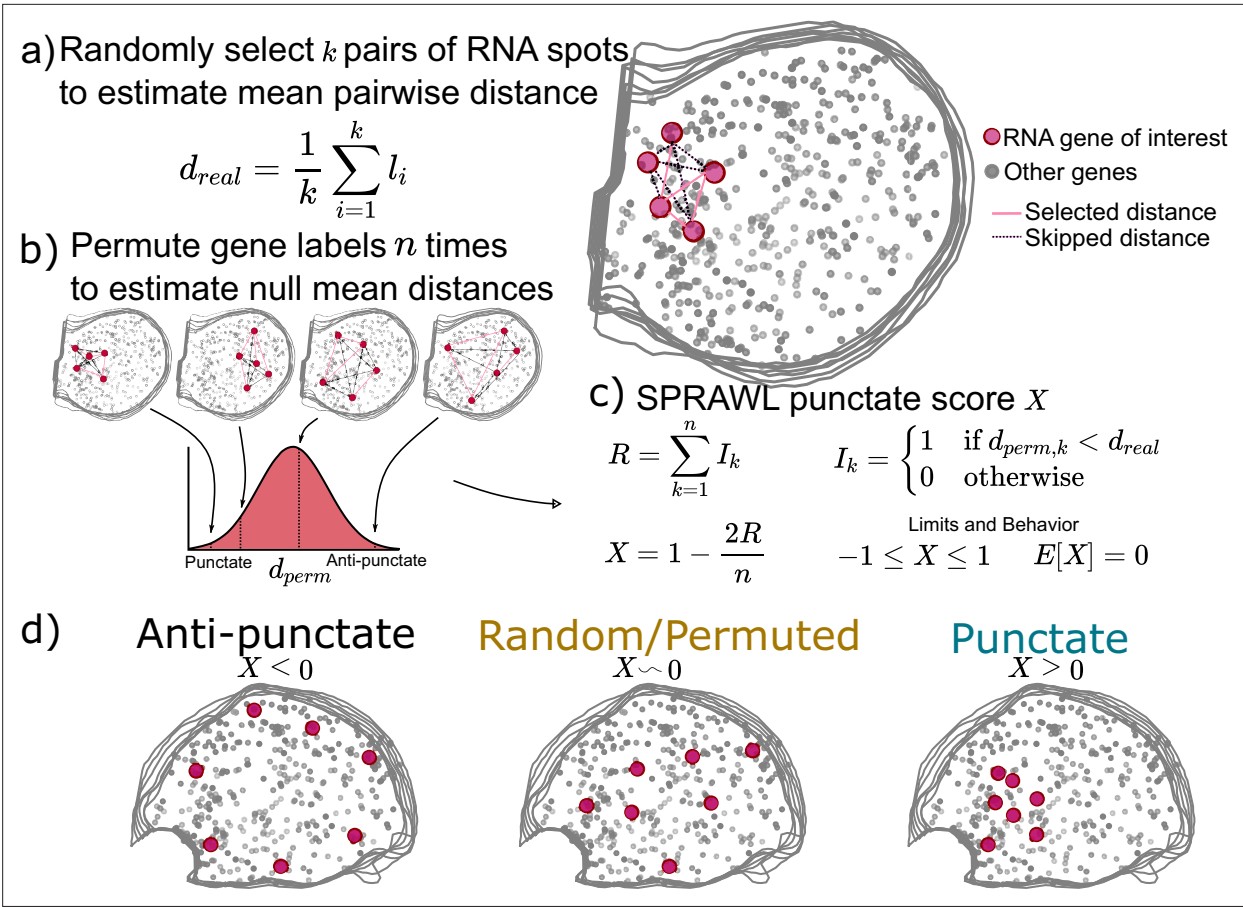

**Figure 2.** Subcellular Patterning Ranked Analysis With Labels (SPRAWL) punctate and radial scores workflow. (**a**) The SPRAWL punctate metric relies on (**b**) permutation testing to create a score (**c**) that represents whether RNA molecules from the gene of interest are closer together than expected by chance. The radial metric is identically calculated, except using average angle instead of distance. The significance of gene-cell-type punctate patterns is calculated using the Lyapunov Central Limit Theorem (CLT) as in the peripheral metric. (**d**) Depictions and interpretation of the SPRAWL punctate metric.

Unlike the peripheral and central metrics, the radial and punctate scores rely on permutation testing to create a null distribution for each gene in each cell. The advantages of permutation testing are that the metrics can be of any complexity, but the disadvantage is the increased compute time in comparison with the simpler rank-based approaches. The permutation-based metrics retain the critical insensitivity to cell size, shape, and orientation present in the rank-based metrics.

## SPRAWL robustly detects subcellular localization in massively multiplexed FISH datasets

The SPRAWL peripheral, central, punctate, and radial metrics described above have been used to analyze spatial datasets comprising a total of 26 experiments over six mice processed by three different research groups and two technologies (*Eng et al., 2019*; *Zhang et al., 2021*; *Vizgen, 2024*; *Liu et al., 2022*). Applying SPRAWL to these datasets revealed: (1) gene/cell-type localization patterns have a high correlation between biological replicates; (2) differential subcellular localization patterns of the same gene in different cell-types; and (3) differential subcellular regulation corresponding with cell-type differential 3' UTR length from associated single-cell RNA sequencing (scRNA-seq) datasets (*Yao et al., 2021*) for 26 genes including *Slc32a1*, *Cxcl14*, *Nxph1*, and *Timp3*.

## SPRAWL detects cell-type specific localization patterns across biological replicates

We applied SPRAWL to the BICCN motor cortex (MOp) (*Zhang et al., 2021*), Vizgen Brainmap, and Vizgen Liver datasets (*Vizgen, 2024*) which each contained either biological or technical replicates. The median SPRAWL gene/cell-type scores were significantly positively correlated between replicates

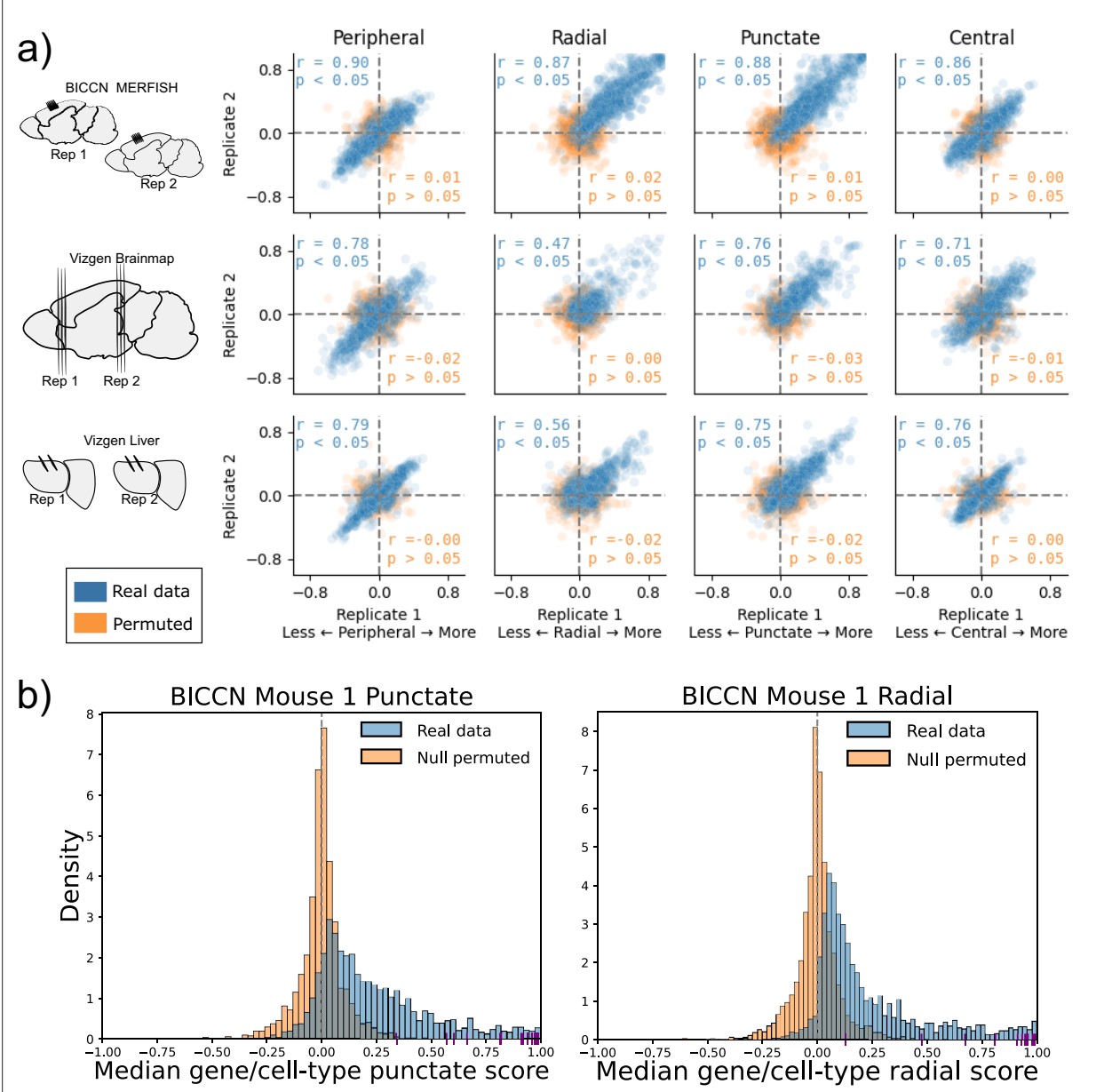

**Figure 3.** Subcellular Patterning Ranked Analysis With Labels (SPRAWL) gene/cell-type scores are highly correlated between biological replicates. (**a**) BICCN MERFISH, Vizgen Brainmap, and Vizgen Liver biological replicates (rows top to bottom) have Pearson correlation coefficients (blue) larger than 0.47 for SPRAWL peripheral, radial, punctate, and central metrics (columns left to right). Randomly permuting gene labels in these datasets eliminates underlying spatial patterning and yields insignificant Pearson correlation coefficients (orange) between biological replicates. Dotted lines indicate zero-valued SPRAWL gene-cell type scores. (**b**) In the motor cortex (MOp) BRAIN Initiative Cell Census Network (BICCN) dataset 87% of gene/cell-type pairs have positive punctate RNA patterning (blue), compared to 50% in the gene-label permuted data (orange). Similarly extreme trends of 95% and 52% are observed for the radial metric. *Cldn5* RNA is consistently highly punctate and radial in all cell-types that express it, depicted by purple x-axis ticks.

The online version of this article includes the following figure supplement(s) for figure 3:

**Figure supplement 1.** Vizgen Liver Showcase scores are highly correlated between replicates.

within all three datasets for all four spatial metrics having significant Pearson correlation with coefficients larger than 0.47 at an alpha level of 0.05 (*Figure 3a*: blue).

Given the observed high pervasiveness of subcellular patterning in all datasets, we tested the specificity of SPRAWL by using permuted data. By permuting the gene-label of the RNA spots in a cell, we create negative control datasets that are known not to have significant spatial patterning. Assuringly, SPRAWL median gene/cell-type scores were not significantly correlated between biological replicates in any permuted dataset (*Figure 3a*: orange). Furthermore, in these negative control datasets, SPRAWL does not call any gene to be significantly localized in any cell-type after correcting for multiple hypothesis testing.

As an additional control for SPRAWL specificity, MERFISH and Vizgen experiments include 'blank-codes' which do not correspond to actual genes and are, therefore, not expected to have significant spatial patterning. In the BICCN MOp dataset, 10 blank-codes were included which SPRAWL determined to be spatially regulated in only the radial and punctate metrics. For the punctate metric, 191 of the 248 unique genes that had statistically significant patterning in at least one cell-type had smaller BH-corrected p-values than the most significant blank-codes. Similarly, for the radial metric, 232 of the 241 unique significant genes had a smaller p-value than the most significant blank-codes. SPRAWL did not identify significant patterning of blank-codes in any cell-type pairings across all replicates for the Vizgen Brainmap and Vizgen Liver datasets. Therefore, adjusting the p-value thresholds to filter out blank-codes would result in the loss of only 57 punctate and nine radial gene significance calls from only one dataset, again supporting SPRAWL's specificity.

To test whether the SPRAWL peripheral score was sensitive to cell-segmentation, we compared SPRAWL before and after mutating the cell boundaries of a dataset (Methods and *Figure 1—figure supplement 1e*). Specifically, the cell boundary locations were computationally shrunk by a factor of 1.25 fold in the x and y direction, discarding spots that fell outside the new boundaries. In both the BICCN MOp and Vizgen Brainmap datasets, a Pearson correlation coefficient of greater than 0.85 was observed between the shrunk and original median gene/cell-type periphery scores. Insensitivity to cell-segmentation is an important feature of a subcellular localization algorithm due to the multitude of approaches and noted difficulties in computational cell segmentation (*Coelho et al., 2009*; *Thomas and John, 2017*; *Vicar et al., 2019*; *Durkee et al., 2021*).

While SPRAWL's specificity can be benchmarked with multiple approaches, estimating SPRAWL's sensitivity on real datasets is confounded by a lack of known true positive subcellular RNA patterning by cell-type. As a proxy for ground-truth, we hypothesized that RNAs encoding proteins with a signal recognition particle (SRP+) would have more centralized patterning than RNAs without (SRP-) due to their known trafficking to the endoplasmic reticulum. Surprisingly, the scores of all SPRAWL metrics were indistinguishably distributed between SRP+ and SRP- genes (*Figure 5—figure supplement 1a*). In an additional approach, we tested whether highly central RNAs were enriched in single-nucleus sequencing (snRNA-seq), compared to scRNA-seq, which was true for only a subset of genes (*Figure 5—figure supplement 1b*). A potential reason for both ground-truth proxies behaving unexpectedly is the nucleus is not necessarily centrally localized and RNAs may not be detectable when protein-bound.

## Cell-type specific subcellular localization is regulated in BICCN MOp replicates

In the MOp dataset, SPRAWL detects hundreds of significantly patterned gene/cell-type groups. The MOp dataset imaged 252 genes through multiplexed barcoding, including 10 negative-control barcodes, and profiled nearly 300,000 cells from the mouse motor cortex (*Zhang et al., 2021*). Biological replicates were present from two mice (*Figure 3a*: top row) with six slices taken from each animal. Conservative filtering of cells and cell-types (see Methods: SPRAWL Filtering) resulted in 220 unique genes and 19 distinct cell-types, with 1999 of 4180 (47.8 %) possible gene/cell-type combinations observed. After BH multiple hypothesis testing corrections over both biological replicates, 1511 (75.6%) gene/cell-type pairs were called significant by the SPRAWL peripheral metric, 1492 (74.6%) by the central metric, 1475 (73.8%) by the radial metric, and 1448 (72.4%) by the punctate metric. Spatial patterning was extensive and consistent between replicates with more than 77.8% of the gene/cell-type pairs having the same direction of effect, positive or negative, between the two replicates. Additionally, 176 of 220 (80%) unique genes were found to be significantly spatially regulated

in at least one cell-type in all metrics, but not necessarily the same cell-type in all metrics. Similarly, all 19 cell-types were observed to be significant with at least one gene in each metric.

## Cell-type specific subcellular localization is regulated in Vizgen Brain replicates

The Vizgen Brainmap dataset contains nine MERFISH experiments from three coronal sections of a mouse brain. Each section contains three adjacent cryotome slices from the same animal that are considered pseudo-biological replicates (*Figure 3a*: middle row). Approximately 70,000 cells and 649 genes, of which 165 were blank-code negative controls, were imaged. Cell-type annotations were not provided for this dataset, and instead, a simple clustering of cells by gene count from the spatial data was performed using Scanpy (*Wolf et al., 2018*) that resulted in 42 cell-type proxies (Methods, Brainmap clustering). Analysis of the three brain slices resulted in 158 (54.7%), and 159 (55.0%), 139 (48.1%), and 156 (54.0%) unique genes significant in at least one cell type for the peripheral, central, radial, and punctate analysis, respectively. For the peripheral metric, 2535 of 2877 (88.1%) gene/cell-type groups present in all three tissue slices had significant Benjamini-Hochberg corrected p-values ($\alpha$=0.05). A similar 87.7% of gene/cell-types were significant according to the centrality metric. For the radial metric, 1194 of 2877 (51.6%) gene/cell-type groups were significant, while the punctate metric identified 2196 of 2877 (76.3%) of the gene/cell-type pairs as significant.

All slices from all sections were pairwise significantly correlated for the peripheral, radial, and punctate metrics with a minimum Pearson correlation coefficient of 0.55. Cell-type SPRAWL correlation results were insensitive to different cell-type clustering parameters (*Figure 3—figure supplement 1*), suggesting that the agreement between biological replicates found by SPRAWL is robust to different granularities of clustering; a desirable trait since cell-type clustering approaches vary widely.

## Cell-type specific subcellular localization is regulated in Vizgen Liver replicates

The Vizgen Liver dataset consists of two mice, each with two replicates for a total of four MERFISH experiments (*Figure 3a*: bottom row). Spatial data was collected on more than 1 million liver cells across all four datasets and 589 distinct genes were imaged, of which 127 were blank-codes. As with the Vizgen Brainmap dataset, no cell-type annotations were provided and naive clustering was performed to generate pseudo-annotations. After filtering out gene/cell-type groups with fewer than 20 cells, SPRAWL detected 112 (29.1%) peripheral, 112 (29.1%) central, 118 (30.6%) radial, and 134 (34.8%) punctate genes significant in at least one cell-type. Median SPRAWL scores per gene/cell-type were highly correlated between the biological replicates with Pearson correlation coefficients of 0.79, 0.56, 0.75, and 0.76 for the peripheral, central, radial, and punctate metrics, respectively. The peripheral metric identified 1399 of 1642 (85.2%) significant gene/cell-type pairs after restricting to median RNA spot count greater than or equal to 5, and presence in both biological replicates. Similar percentages of 85.1%, 51.4%, and 77.4% of gene/cell-type pairs were found to be significantly patterned in the central, radial, and punctate metrics.

## Significant SPRAWL punctate and radial scores are highly skewed towards aggregation

Over 99% of the significant gene/cell-type groups have positive (X>0) radial and punctate scores, revealing a significant and general tendency of RNAs to colocalize with other RNAs of the same gene both by euclidean distance (punctate metric), and angular dispersion (radial metric). In comparison, the SPRAWL peripheral metric in the BICCN MOp dataset identifies 56.1% of significant gene/cell-type pairs as more positively peripheral (X>0) and the remaining 43.9% are anti-peripheral (X<0). Similarly, the SPRAWL central metric identifies 45.2% of significantly positive scoring gene/cell-type pairs. Empirical CDF plots of SPRAWL metric scores provide an alternate view for the same phenomenon (*Figure 1—figure supplement 1a, b*). Additionally, null simulated datasets did not have a bias towards positive radial or punctate scoring (*Figure 3a* orange).

SPRAWL detects 112 of 252 genes (44.4%) as globally positively punctate and radial in all cell-types which express them including extreme genes, such as Claudin 5 (*Cldn5*) which has a median SPRAWL punctate and radial score of 0.85 and 0.84, respectively (*Figure 3b* purple ticks) as well as VEGFR-1 (*Flt1*) which has a median SPRAWL punctate and radial score of 0.83 for both metrics (*Figure 3b*).

*Cldn5* protein product is the primary integral membrane protein component of tight junctions in mouse brain and knockouts result in postnatal death (*Nitta et al., 2003*). *Flt1* is a transmembrane tyrosine kinase receptor that binds vascular endothelial growth factor (VEGFR) and also has a shortened alternative soluble protein isoform (*Shibuya et al., 1990*; *Jin et al., 2012*). The consistent positive punctate and radial scores of *Flt1*, and lack of differential localization patterns, could indicate that either only one isoform of *Flt1* is expressed across all cell-types, or that the two mRNA isoforms are alternatively expressed but do not have differential subcellular localization patterns. It is currently not known in the literature whether *Cldn5* or *Flt1* RNA localization is regulated, but a followup targeted FISH experiment could be insightful. We note that imaging errors resulting in calling a single RNA molecule as two nearby molecules could be artificially inflating the radial and punctate scores leading to more significant calls.

## SPRAWL detects genes with opposite and cell-type dependent RNA localization

We defined opposite-directionality genes as those that have the pattern of being significantly positively scoring in one cell-type, while being significantly negatively scoring in another cell-type for the same metric, such as peripheral vs. anti-peripheral. Significant spatial patterning of a gene in only a subset of cell-types suggests differences in either cis or trans-acting regulatory factors. For the BICCN dataset out of 252 genes, 92 (36%) peripheral, 96 (38%) central, 2 (1%) radial, and 10 (4%) punctate genes are opposite-directionality (Supplemental Table 1 in *Supplementary file 1*). We define an additional class of genes as cell-type dependent, but not opposite-directionality patterning. These genes are significant in at least one cell-type, but insignificantly localized in at least one other cell-type and account for approximately 55% of genes in peripheral and central metrics, and 20% for the radial and punctate metrics across all datasets. SPRAWL's ability to detect cell-type specific regulation of subcellular patterning generates testable hypotheses for follow-up analysis and experimentation. A computationally tractable hypothesis of interest inspired by the known presence of 'zip code' elements, is whether there exist general correlations between 3' UTR isoform and localization across cell-types.

## Subcellular RNA localization is enriched for correlations with 3' UTR length

Alternative 3' UTRs and splice isoforms are known to result in differential mRNA localization (*Kislauskis et al., 1994*). Inclusion or exclusion of specific sequence elements can disrupt RNA binding proteins (RBPs) from binding and localizing the transcript. RBPs that have been identified as controlling transcript localization can have cell-type specific expression, including at the isoform level (*Yisraeli, 2005*; *Müller-McNicoll and Neugebauer, 2013*; *Hentze et al., 2018*). Examples of such RBPs include members of the RNA-transport granule (*Kanai et al., 2004*), providing a model for why RNAs may be cell type specifically localized as a function of their isoform. Conversely, differential localization of the same isoform can occur if the trans-acting localization factor is differentially expressed in different cell-types.

We coupled a recent statistical method to measure 3' UTR length called the ReadZS (*Meyer et al., 2021*) with SPRAWL to identify genes with spatial localization correlated with 3' UTR regulation (*Booeshaghi et al., 2021*). We used ReadZS to statistically quantify 3' UTR lengths at single-cell resolution, and then computed the median ReadZS score by cell-type and gene on cell-type-matched 10Xv3 scRNA-seq datasets from the BICCN consortium (*BRAIN Initiative Cell Census Network (BICCN), 2021*). Spatial localization SPRAWL scores and ReadZS 3' UTR lengths were correlated by gene/cell-type (*Figure 4a*). Twenty-six genes were detected as having significant SPRAWL/ReadZS correlation after BH multiple hypothesis correction at an FDR level of 0.05, a twofold enrichment compared to what is expected by chance (see Methods: Correlation analysis between SPRAWL and ReadZS). No significant gene/metric pairs were detected from the CZB mouse kidney/liver dataset which was the only other dataset with matched scRNA-seq. The lack of significant correlations between the SPRAWL metric score and 3' UTR length in this dataset could be due to multiple factors, including this dataset having fewer coarser cell-type definitions.

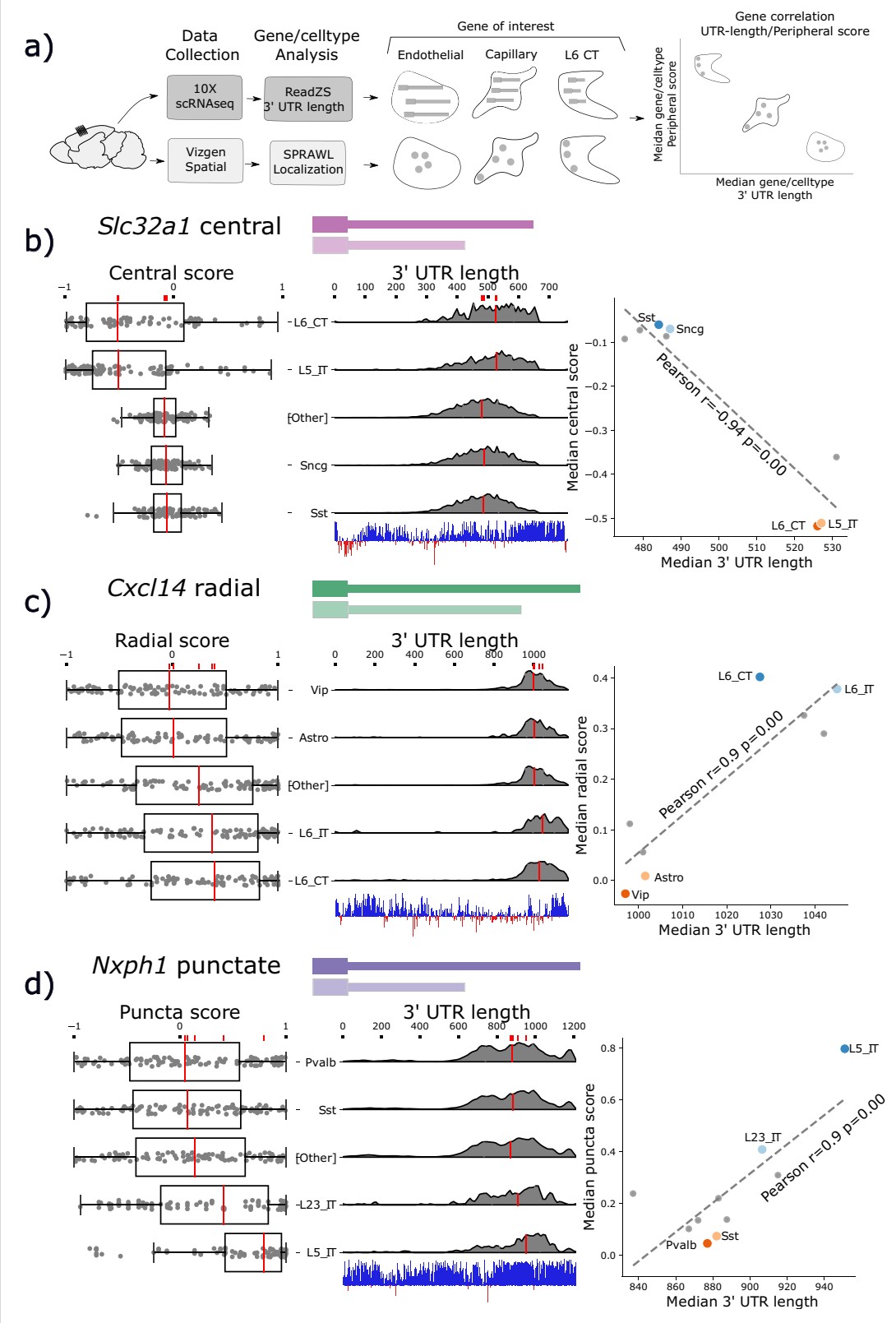

**Figure 4.** Subcellular Patterning Ranked Analysis With Labels (SPRAWL) spatial scores and 3' Untranslated Region (UTR) length are significantly correlated for a subset of genes. (**a**) Workflow to calculate median 3' UTR length and spatial score per gene/cell-type. (**b**) *Slc32a1* median centrality, (**c**) *Cxcl14* radial, and (**d**) *Nxph1* punctate SPRAWL scores from the BRAIN Initiative Cell Census Network (BICCN) MERFISH dataset correlate significantly with 3' UTR length determined from 10 X scRNA-seq data by ReadZS. The left-column boxplots show individual SPRAWL cell scores as overlaid dots.

*Figure 4 continued on next page*

*Figure 4 continued*

The cell-types are sorted by increasing median score marked in red. The two cell-types with the highest and lowest median SPRAWL scores are plotted individually while the remaining cell-types are collapsed into the 'Other' category. Gene/cell examples are shown to the left of the boxplots for each extreme cell-type group. The density plots in the middle column show estimated 3' UTR lengths for each read mapping within the annotated 3' UTR, stratified by cell-type. Lengths were approximated as the distance between the annotated start of the 3' UTR and the median read-mapping position. Each density plot is normalized by cell-type to show relative shifts in 3' UTR length with median lengths depicted with red lines. The scatterplots show the significant correlations between the median SPRAWL score and the median 3' UTR length. The two cell-types with the highest, and the two with the lowest SPRAWL median scores are highlighted.

The online version of this article includes the following figure supplement(s) for figure 4:

**Figure supplement 1.** ReadZS detects Tabula Sapiens Lung differential 3' Untranslated Region (UTR) length *TIMP3* and decreases in *Timp3* expression throughout culture.

**Figure supplement 2.** Computationally predicted miRNA binding sites in the 3' Untranslated Regions (UTRs) of Slc32a1, Cxcl4, and Nxph1 and additional 3' UTR correlated genes.

## *Slc32a1*, *Cxcl14*, and *Nxph1* 3' UTR length predicts sub-cellular localization

SPRAWL detects 26 unique genes and 84 pairs of gene/metric combinations (i.e. gene1/peripheral, gene1/radial) with significant correlations to that gene's 3' UTR length. From this list, *Slc32a1*, *Cxcl14*, and *Nxph1* were selected as representatives of the central, radial, and punctate metrics, respectively. All have significant evidence for cell-type differential expression of un-annotated 3' UTRs and an unusually high degree of 3' UTR conservation. *Figure 4* depicts the SPRAWL scores and predicted 3' UTR lengths for *Slc32a1*, *Cxcl14*, and *Nxph1* in multiple cell-types. Representative low and high-scoring cells for each gene/cell-type pair were chosen randomly after filtering for SPRAWL scores less than –0.2 and greater than 0.2, respectively, having 5 or more RNA spots of the gene of interest.

*Slc32a1*, synonymously *VIAAT* or *VGAT*, is a marker of GABAergic neurons and was found to be differentially central by cell-type (*Figure 4b*). Slc32a1 is an integral membrane protein residing in synaptic vesicles where it uptakes glycine and gamma-aminobutyric acid (GABA) (*Gasnier, 2004*). *Slc32a1* is currently annotated to have two exons in the UCSC genome browser mm39 (*Lee et al., 2022*), but was at one point thought to have three exons and exhibit alternative splicing near the 3' UTR without known biological significance (*Ebihara et al., 2003*). SPRAWL central score and ReadZS have a significant correlation (Pearson $R=-0.94$, corrected $p \ll 0.05$). Differential central localization of *Slc32a1* RNA between cell-types is of potential interest due to the protein product's known role of localizing to synaptic vesicles in neurons which would yield the highly non-central distribution observed in the L6 CT and L5 IT neuronal cell-types.

*Cxcl14* 3' UTR length and SPRAWL radial score were significantly correlated (Pearson $R=0.9$ corrected $p \ll 0.05$); cell-types with longer 3' UTRs have increasingly extreme radial clustering, while the unannotated shorter 3' UTRs have middling SPRAWL non-radial scores near zero. Only one *Cxcl14* 3' UTR isoform is annotated, but ReadZS analysis predicts a decrease in length of about 600 bps (*Figure 4c*; *Bässler et al., 2001*). The protein product of *Chemokine (C-X-C motif) ligand 14*, *Cxcl14* or *BRAK*, is a small chemokine of length 99 residues in mouse and 111 in humans, and was originally found to be highly expressed in breast and kidney (*Hromas et al., 1999*). *Cxcl14* is constitutively expressed in skin and keratinocytes and is a potent leukocyte recruitment factor (*Westrich et al., 2020*) but has also more recently been observed as constitutively expressed throughout multiple brain regions where one of its functions is to regulate synaptic transmission (*Banisadr et al., 2011*). According to the MERFISH dataset, *Cxcl14* was lowly but consistently expressed with the full-length 3' annotated UTR in 429 L6 neurons with a median of 5 spots per cell while having higher expression in Vip-cells and astrocytes where a slightly shorter 3' UTR was expressed. We hypothesize that *Cxcl14* has differential 3' UTR usage associated with differential expression across these cell-types and that the novel short 3' UTR is less radially clustered than the annotated full-length 3' UTR.

*Nxph1*, neurexophilin-1, is a ligand of *Nrxn1* and is expressed in inhibitory neurons (*Born et al., 2014*). The punctate SPRAWL score of *Nxph1* is positively correlated with 3' UTR length (Pearson $R=0.9$, corrected $p \ll 0.05$ *Figure 4d*). *Nxph1* is a secreted protein that binds to multiple splice isoforms of *Nrxn1* at synapses with varying specificity (*Wilson et al., 2019*). To our knowledge, neither differential 3' UTR lengths nor differential subcellular localization patterns have been previously

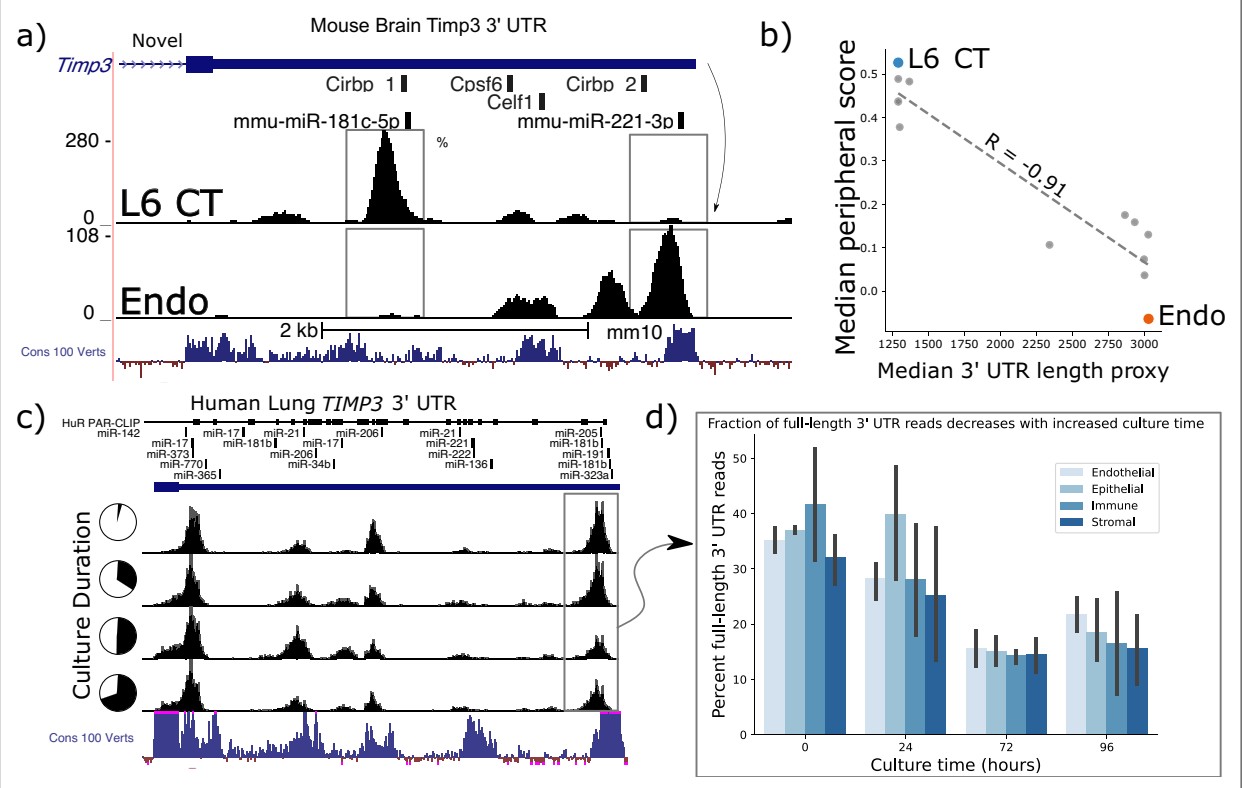

**Figure 5.** *Timp3* alternative peripheral localization across motor cortex (MOp) cell types is statistically correlated with ReadZs differences in 3′ Untranslated Region (UTR) length. (**a**) ReadZs detects two major alternative 3′ UTRs in mouse *Timp3* from 10 X scRNA-seq which correspond to miR-181c-5p and miR-221–3 p binding sites. Reads from L6 critical threshold (CT) cells predominantly map to a novel upstream shortened 3′ UTR while endothelial cells primarily express the longer annotated 3′ UTR. The UCSC genome browser placental animal sequence conservation shows highly conserved regions in blue. Fisher's exact test was highly significant between the two peaks denoted by the dotted lines between the two cell types. (**b**) *Timp3* mean periphery score is significantly correlated with *Timp3* median ReadZs score across MOp cell-types with Pearson correlation coefficient of −0.91 and p<<0.05. (**c**) Fraction of *TIMP3* RNA full-length 3′ UTR reads, gray box, and (**d**) bar plots, decreases during human lung tissue culture.

The online version of this article includes the following figure supplement(s) for figure 5:

**Figure supplement 1.** Subcellular Patterning Ranked Analysis With Labels (SPRAWL) scores do not correlate with the presence of signal recognition peptide, but do correlate with nuclear enrichment.

described for *Nxph1*, although dendritic targeting by 3′ UTRs of other proteins, such as *CaMKII*, has been identified (*Mayford et al., 1996*).

All three genes, *Slc32a1*, *Cxcl14*, and *Nxph1*, have predicted miRNA binding sites tiling their 3′ UTRs suggesting possible mechanisms of differential 3′ UTR post-transcriptional selection and regulation (*Figure 4—figure supplement 2a*). We show an additional three genes with correlated spatial and 3′ UTR length show similar patterns (*Figure 4—figure supplement 2b*).

### *Timp3* 3′ UTR length predicts peripheral localization

In the BICCN data, *Timp3* has the largest observed variation in estimated 3′ UTR length between cell-types, with the most divergent read-buildup between layer-6 corticothalamic (L6 CT) and endothelial cells reflecting at least two dominant 3′ UTRs differing in length by >2 kilobases (*Figure 5a*). These 3′ UTR read densities were consistent across mouse biological sequencing replicates within 10 X scRNA-seq experiments. Only one UTR is annotated, though a gene antisense to *Timp3*, *Sync3* on the minus strand, overlaps its transcriptional radius. We are confident that observed reads can be confidently attributed to *Timp3* as *Sync3*'s nearest exon is ~5 kb from *Timp3*'s UTR and plus-strand mapping reads alone were analyzed.

Timp3 is a secreted matrix metalloprotease inhibitor that has been implicated in multiple diseases ranging from cardiomyopathies to macular dystrophies (*Weber et al., 1994*; *Schrimpf et al., 2012*), but subcellular RNA localization patterns have not been reported. Elevated *Timp3* gene expression

(*Capone et al., 2016*) is linked to compromised cerebral blood flow, and the RNA is experimentally validated to be a target of microRNA (miRNA) regulation (*Fiorentino et al., 2013*). We observe *Timp3* RNA to be significantly peripheral in L6 IT neurons; while being insignificantly peripherally localized in Sst cells. SPRAWL and ReadZS 3' UTR scores had a significant negative correlation of $R=-0.68$ and $p<<0.05$ Pearson BH-corrected p-value. *Timp3's* longer, annotated 3' UTR isoform is expressed in cell-types with significantly less peripheral localization as compared to shorter unannotated isoforms (*Figure 5b*).

We studied whether *Timp3's* 3' UTR length was more globally regulated in endothelial and other cell-types through scRNA-seq and in different biological contexts in both mouse and human datasets and extended the analysis to include *Timp2*. Mouse and human *Timp3* have a 96.2% amino acid sequence similarity with mouse and human *Timp2* having an even higher 98.2% sequence identity. ReadZS also detected statistically significant *Timp3* 3' UTR length shifts between cell-types from the Tabula Sapiens consortium *Jones et al., 2022* in the lung and other tissues (*Figure 4—figure supplement 1a*). Furthermore, we found both *Timp2* and *Timp3* UTR lengths to be regulated in lung tissue slices across endothelial, epithelial, immune, and stroma cell-type compartments (*Figure 5c and d*). Since SPRAWL identified a highly negative correlation between *Timp3* peripheral subcellular localization and 3' UTR length, and since *Timp3* 3' UTRs become shorter during lung culture, the subcellular localization of *Timp3* is predicted to shift to a more peripheral distribution during the lung culture. In conjunction with 3' UTR length shortening, gene expression of *Timp3* decreases over this time course in all cell-type groups (*Figure 4—figure supplement 1b*).

Both mouse and human *Timp3* show high conservation within its 3' UTR. Conservation is particularly high near the two dominant alternative 3' UTR regions (*Figure 5a and c*: Cons 100 Verts track), all but one of which are un-annotated. These regions could contain alternative end processing or regulatory sequences. In the case of mouse *Timp3*, this includes annotated binding sites for miR-181c-5p and miR-221–3 p and RBPs *Cirbp*, *Cpsf6*, and *Celf1* (*Figure 5a*). The 3' UTR isoforms differentially include these regions, releasing the shorter isoforms from regulatory pressures by more distal elements, including the experimentally validated miR-21 that binds in the 3' UTR of human *Timp3* (*Hu et al., 2016*). In this study, the authors found that high expression of miR-21 led to repression of *Timp3* and pathogenic activation of angiogenesis.

Together, we hypothesize that Timp3 may have both secreted and non-secreted isoforms, with a precedent set by *Cd47* (*Berkovits and Mayr, 2015*). Furthermore, we hypothesize that this regulation is controlled by alternative 3' UTR isoform lengths that determine subcellular RNA localization through interaction with RBPs and microRNAs that specifically bind the longer isoform. This example illustrates the power of SPRAWL for unsupervised discovery of subcellular localization and its integration with isoform-resolved, annotation-free analysis of scRNA-seq to generate testable biological hypotheses regarding isoform-specific regulation and function.

## Human brain pericyte cell culture shows differential temporal Timp3 3' UTR usage

Motivated by the findings that (1) mouse brain cell-types expressing shorter *Timp3* 3' UTR isoforms were correlated with increasingly peripherally localized *Timp3* RNA (*Figure 5b*), and (2) *Timp3* 3' UTR lengths decrease throughout human lung slice culture (*Figure 5c and d*), we hypothesized that Timp3 protein secretion would be sensitive to RNA localization and/or 3' UTR length. We tested this hypothesis using a human brain pericyte cell-line known to express Timp3 protein. The pericytes were cultured over 5 d with supernatant samples collected at 6, 24, 48, and 72 after plating. At each timepoint, the number of cells, total extracellular protein concentration (BCA), extracellular Timp3 protein (ELISA), and *Timp3* RNA (qPCR) were measured (*Figure 6a*).

We observed that the rate of per-cell Timp3 protein secretion, as measured by ELISA, does not significantly change throughout culture time, averaging 350 Timp3 protein molecules per-cell per-hour. The approximately 15- hr half-life of Timp3 protein in cell culture (*Mao et al., 2021*) was taken into account when making these calculations (Methods Timp3 protein production estimation). However, total extracellular protein per-cell slightly decreased from 6 to 24 hr of cell culture as measured by BCA (*Figure 6b*). Taken together these findings suggest that Timp3 protein production is not variable during cell culture.

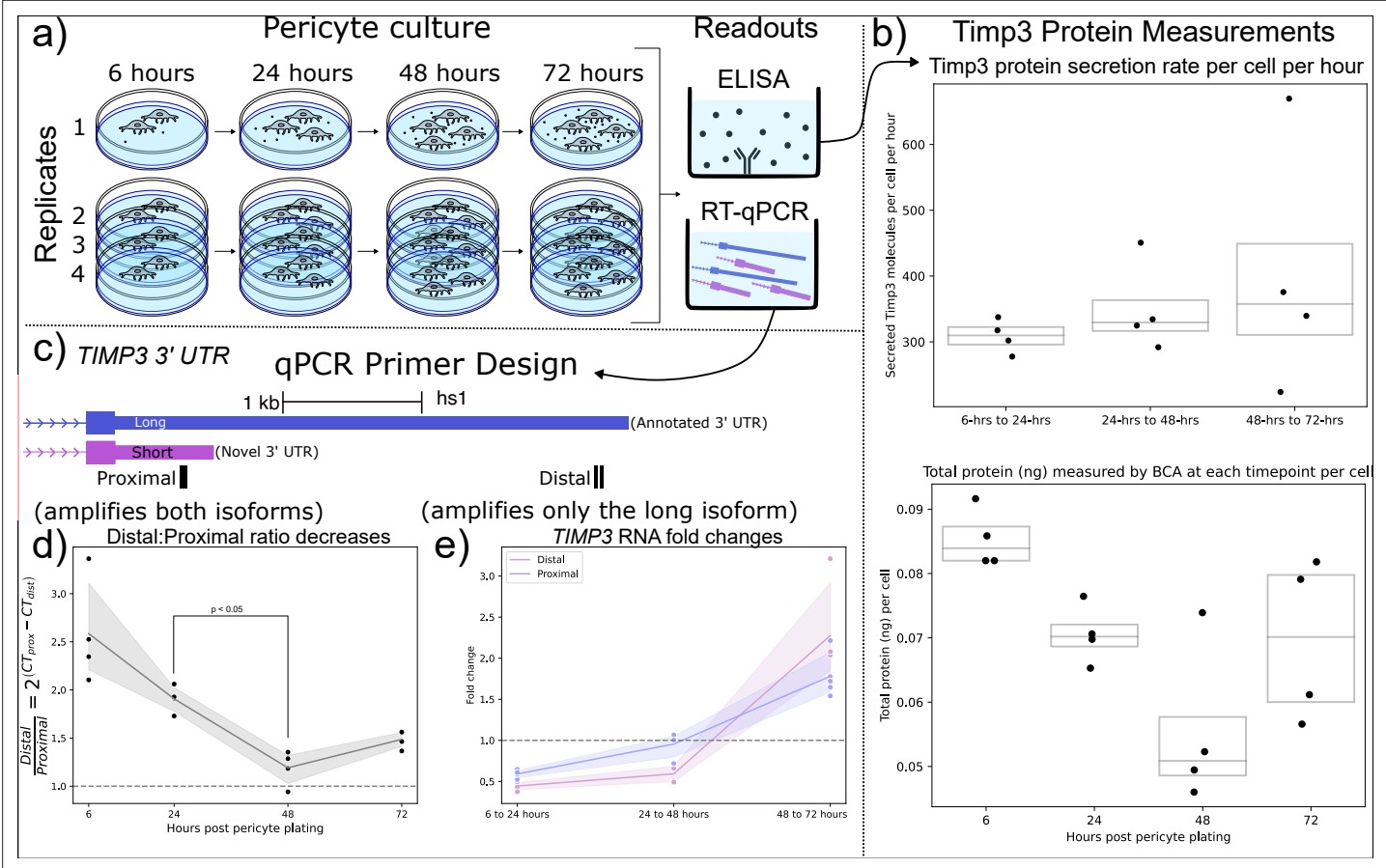

**Figure 6.** Shorter *TIMP3* 3' Untranslated Regions (UTRs) become relatively more abundant in pericyte cell culture while *TIMP3* protein production remains stable. (**a**) Experimental setup for human pericyte cell culture with reverse-transcriptase quantitative PCR (RT-qPCR) and extracellular *TIMP3* protein ELISA readouts at four-timepoints. (**b**) *TIMP3* protein secretion per cell per hour does not significantly change throughout culture time, even though the total protein measured by BCA does change. (**c**) qPCR experiment design with proximal and distal qPCR primers to distinguish long and short 3' UTR isoforms. The proximal qPCR primer can detect both long and short isoforms while the distal primer can only amplify the long 3' UTR. (**d**) The ratio of distal to proximal primer-template abundances significantly decreases throughout culture time, implying increased usage of the short *TIMP3* 3' UTR compared to the long isoform. (**e**) *TIMP3* 3' UTR abundance, normalized by 18 s housekeeper abundance, fluctuates from halving to doubling between culture timepoints for both distal and proximal primers.

The online version of this article includes the following figure supplement(s) for figure 6:

**Figure supplement 1.** qPCR primer efficiencies for Timp3 3' Untranslated Region (UTR) were estimated by using twofold cDNA dilutions of the same 6 hr timepoint sample.

From the previous human lung culture experiment (*Figure 5c*), we hypothesized that the abundance of shortened 3' UTRs of Timp3 would increase relative to the canonical full-length isoform throughout pericyte cell culture. To test this hypothesis, *Timp3* short and long 3' UTR abundance were estimated using proximal and distal qPCR primers. The proximal qPCR primer pair is designed to amplify both full-length canonical and un-annotated shortened 3' UTR *Timp3* templates. The distal qPCR primer pair, however, can only amplify the full-length isoform (*Figure 6c*). In support of our hypothesis, we observe the ratio of *Timp3* distal to proximal RNA abundance significantly decreased from 24 to 48 hr by a factor of 1.5 X (*Figure 6d*).

Additionally, *Timp3* 3' UTR expression decreased by half between 6 and 24 hr before doubling between 48 and 72 hr as measured by both proximal and distal qPCR primers (*Figure 6e*). The large fluctuations in Timp3 expression relative to multiple house-keeping genes are noteworthy since the Timp3 protein production levels remained constant throughout the experiment. This observation may suggest post-transcriptional or post-translational regulation. In conclusion, transcripts of *Timp3* with

the proximal 3' UTR region increased in relation to the distal region during pericyte culture, which is in agreement with our hypothesis from the human lung culture model.

## Discussion

Highly multiplexed spatial transcriptomics datasets are becoming increasingly available, but analysis tools have overwhelmingly focused on localizing cell-types within tissue, rather than RNA within cells. SPRAWL addresses this need as a novel non-parametric approach for unbiased detection of subcellular RNA localization patterns. In this study, SPRAWL provides evidence for (1) highly consistent RNA patterning across biological replicates, (2) abundant cell-type specific RNA localization, and (3) differential patterning dependent on 3' UTR isoform.

We show that SPRAWL has perfect specificity when benchmarked on simulated negative control datasets, yet identifies thousands of significant genes with extreme RNA localization patterns by cell-type in real datasets. The simplicity of the SPRAWL score facilitates integration with other datasets and tools for follow-up computational studies. We've been able to illustrate this concept by leveraging existing scRNA-seq datasets and the ReadZS tool (*Meyer et al., 2022*) to find genes with correlated patterning and 3' UTR usage. Additionally, SPRAWL results can motivate experimental studies that detect novel biology as we've shown by identifying shifting *Timp3* 3' UTR isoform usage in a pericyte culture experiment.

SPRAWL prioritizes potentially functionally important isoform expression for further study such as *Timp3, Slc32a1, Cxcl14*, and *Nxph1* which have significant spatial and 3' UTR-usage correlation between cell-types. SPRAWL generates testable hypotheses of cis-regulatory elements that alter RNA localization which is of high interest because in mice and humans, more than 96% of genes are alternatively spliced and UTR regulation is pervasive but poorly annotated (*Olivieri et al., 2021*; *Olivieri et al., 2022*).

The localization scores generated by SPRAWL are versatile and can be computed for proteins rather than RNA. In fact, trans-regulated spatial events can be detected in future work by applying SPRAWL to subcellular protein localization datasets generated by tools such as CODEX (*Black et al., 2021*) or MIBI (*Keren et al., 2019*). Furthermore, the SPRAWL framework can be used to implement different measures of subcellular localization. Some but not all statistically significant patterns detected by SPRAWL are 'striking to the human eye,' which has implications for whether human-guided or statistical-guided inferences are preferred and which are more biologically meaningful.

The importance of the correlation between SPRAWL subcellular localization and isoform expression, including *Timp3*, *Slc32a1, Cxcl14,* and *Nxph1*, was minimally explored in this work. Still, we hypothesize a causal link between 3' UTR regulation, localization, and potential protein function, as was observed for *Actb*, which could guide future experimental efforts, as well as help pinpoint cell-type specific functions. Our in vitro human pericyte cell culture experiment, for example, showed that pericytes are utilizing a previously unknown shortened *Timp3* 3' UTR in addition to the full-length isoform. Furthermore, a shift towards more shortened 3' UTR usage occurs during pericyte cell culture; a result that mirrors SPRAWL findings in human lung tissue.

Sampling a handful of tissues and cell types, SPRAWL identified tens of RNA species with subcellular localization related to cell type. Many technical limitations suggest that this number is a significant underestimate: for one, MERFISH-based approaches require probes to be pre-specified, and thus they (a) aggregate isoforms, confounding cases where two co-expressed isoforms have dramatically different localization patterns; (b) miss isoforms that lack sequence contained in the probe set measurements. Furthermore, single-cell sequencing technology and analysis may be under-ascertaining RNA expression due to (i) sampling depth; (ii) poly-A capture bias, and (iii) a dearth of computational algorithms to analyze isoform-specific differences. Through the ReadZS we have collapsed UTR variation to a single scalar value (*Meyer et al., 2021*; *Chaung et al., 2022*; *Olivieri et al., 2022*) but we have not explored correlations with RNA splicing or other sequence variants, a topic of further research. Our findings support a model where 3' UTR regulation at the nucleotide level controls localization through function. If this is true, imaging-based technology like MERFISH will have limited power over discovery and in situ sequencing may be a preferred approach. Together, this suggests that isoform-specific localization may be widespread and confer functions that should be tested in future computational and experimental work.

SPRAWL provides an estimate of the pervasiveness of cell-type and 3' UTR-regulated RNA localization. Limitations of the study include possible confounding by technical artifacts from probe hybridization, improper cell-segmentation, and bias in the gene panel selected for imaging. Additionally, our decision not to use nuclei boundaries blinds us to situations where an RNA may be highly peripheral, but still within the cell nucleus. This could mean UTR peripherality is confounded with the dynamis of export, including transcription at the nuclear periphery. We have attempted to address these potential artifacts through hundreds of thousands of observations and by permutation where possible. Additionally, computationally shrinking cell-boundaries resulted in only minimal changes in SPRAWL scores. Future work on novel datasets using different segmentation approaches will provide further confidence that SPRAWL detects biologically relevant patterns. We believe the current implementation of SPRAWL is conservative and likely misses patterns due to optical crowding and low-abundance gene expression.

There exists no directly competing method to SPRAWL which is able to leverage highly multiplexed imaging datasets, requiring only RNA spot locations, cell-boundary estimates, and gene identity of each RNA spot. Many current software approaches aim to discretize RNA patterning into subcompartments and rely on co-stains which are not guaranteed to be present in every dataset. Other approaches use statistically opaque machine-learning-based classifiers to assign RNA spots to pre-specified patterns (*Mah et al., 2022*). As spatial transcriptomics methods are commercialized and become more accessible, increasing numbers of public datasets will become available and can be processed by SPRAWL regardless of the tissue or study design.

# Materials and methods

## Key resources table

| Reagent type (species) or resource | Designation | Source or reference | Identifiers | Additional information |
|---|---|---|---|---|
| Software, algorithm | SPRAWL | This paper, *Bierman, 2024* | https://github.com/salzman-lab/SPRAWL | |
| Cell line (Homo-sapiens) | Human brain vascular pericytes | Sciencell | #1200 | |
| Sequence-based reagent | Proximal_primer_1_fwd | This paper | Timp3 qPCR primer | GGGAACTATCCTCCTGGCCC |
| Sequence-based reagent | Proximal_primer_1_rev | This paper | Timp3 qPCR primer | TTCTGGCATGGCACCAGAAAT |
| Sequence-based reagent | Proximal_primer_2_fwd | This paper | Timp3 qPCR primer | AGGTCTATGCTGTCATATGGGGT |
| Sequence-based reagent | Proximal_primer_2_rev | This paper | Timp3 qPCR primer | TGGGGCCAGGAGGATAGTTC |
| Sequence-based reagent | Distall_primer_1_fwd | This paper | Timp3 qPCR primer | AATTGGCTCTTTGGAGGCGA |
| Sequence-based reagent | Distal_primer_1_rev | This paper | Timp3 qPCR primer | GCGGATGCTGGGAGAATCTA |
| Sequence-based reagent | Distal_primer_2_fwd | This paper | Timp3 qPCR primer | TAGCCAGTCTGCTGTCCTGA |
| Sequence-based reagent | Distal_primer_2_rev | This paper | Timp3 qPCR primer | GGGTTCGAGATCTCTTGTTGG |
| Commercial assay or kit | qPCR Kit | BioRad | SsoAdvanced Universal supermix | |
| Commercial assay or kit | Human TIMP-3 ELISA Kit | Invitrogen | # EH458RB | |

## SPRAWL input data and preprocessing

SPRAWL takes as input processed datasets from MERFISH, Vizgen, and SeqFISH + requiring cell-boundary and RNA spot x,y, and gene label information. For MERFISH and Vizgen, this data is the product of applying MERlin (*Emanuel and Babcock, 2020*) on the raw MERFISH microscopy images

to align the images between sequencing rounds, call RNA spots, and perform cell segmentation using a seeded watershed approach described in a prior MERFISH work (*Moffitt et al., 2018*). SeqFISH + utilizes a similar approach to identify and decode RNA spots, but then simply defines the cell boundary as the convex hull around all points (*Eng et al., 2019*).

The MERFISH primary mouse cortex (MOp) dataset has 258 genes from coronal slices of the MOp from two mice as biological replicates (*Zhang et al., 2020*). Each mouse had six MERFISH experiments with 5–6 10- um sections processed together on the same coverslip. Each mouse had 32 total sections. Each 10- um thick section had seven optical layers spaced 1.5 microns apart. The MERFISH brain MOp processed datasets include multiple z-slices for each cell. The data was downloaded from https://download.brainimagelibrary.org/cf/1c/cf1c1a431ef8d021/processed_data/.

The SeqFISH + dataset imaged 913 cells and 10,000 genes in the mouse cortex at a single z-slice (*Eng et al., 2019*). The authors assigned each cell to one of twenty-six different cell-type annotations such as Endothelial, Interneuron, Astrocyte, etc. The dataset was downloaded from https://github.com/CaiGroup/seqFISH-PLUS/ (*Cai et al., 2019*).

The Vizgen MERFISH Mouse Brain Map (BrainMap) is a dataset of 649 total genes which include canonical brain cell type markers, GPCRs, and RTKs from a single mouse brain (*Vizgen, 2024*). Three full coronal sections were processed along the rostral-caudal axis. Additionally, for each section, three adjacent slices were used as biological replicates with the underlying assumption that adjacent slices in the mouse brain have high similarities in cell-type composition and spatial organization. Each of the nine imaging datasets contain seven optical layers spaced 1.5 microns apart. Data is publicly available https://console.cloud.google.com/marketplace/product/gcp-public-data-vizgen/vizgen-mouse-brain-map.

The Vizgen MERFISH Liver showcase contained 2 mouse liver samples each with two MERFISH experiments imaging 347 genes and over one million cells (*Vizgen, 2024*). Cell-type annotations were not provided and instead, cell-type proxies were determined by clustering the cells based on the MERFISH-determined RNA composition of each cell (Methods: Vizgen Brainmap and Liver showcase clustering to produce cell-type proxies). The dataset contains seven optical layers spaced 1.5 microns apart and data is publicly available from https://info.vizgen.com/mouse-liver-data?submissionGuid=832a9f61-22d3-44c1-a2cf-838c166d9ac5.

The CZB kidney/liver dataset contained a single mouse kidney and liver sample that were imaged using the Vizgen platform to detect the same panel of 307 genes in ~57,000 cells in the kidney and ~16,000 in the liver (*Liu et al., 2022*). https://figshare.com/projects/MERFISH_mouse_comparison_study/134213.

We have specified a simple HDF5 format to standardize the different data sources. In brief, data is stored in a cell-centric manner, consolidating RNA spots and cell boundaries into the same object. This flexible format is described in detail in the GitHub repository, (copy archived at *Bierman, 2024*) and includes vignettes with example datasets. For MERFISH and Vizgen datasets, the RNA spots and cell boundaries were assigned locations in a global coordinate, but lacked cell assignments for each RNA spot. We have written simple and fast scripts to make these assignments using the python Rtree and shapely (*Gillies et al., 2007*) packages. The GitHub repository includes the next flow pipelines used to transform the downloaded datasets to this HDF5 format.

## SPRAWL methodology

SPRAWL preprocesses spatial datasets into a standardized HDF5 file that contains cell boundary, cell-type, and RNA location information generated from MERFISH/Vizgen and SeqFISH + datasets (*Figure 1a*). Next per-gene/per-cells are calculated. For the peripheral metric, all RNA spots are ranked based on their minimum distance to the cell boundary (*Figure 1b*), and then their means are used to generate a gene/cell-type score and p-value (*Figure 1c*). Scores near 1 indicate a gene is highly peripheral in a cell-type, while scores near –1 indicate a pattern of RNA molecules far from the cell-boundary. Intuitively, if a gene is not significant it will not be close or far from the cell boundary and its peripheral score will be near 0, and its p-value will be insignificant. The centrality metric is conceptually similar, where ranking is determined by minimum distance to the cell centroid and positive values indicate unexpectedly centrally-biased distributions. Empirically, the centrality and peripherality metrics are anti-correlated (*Figure 1—figure supplement 1b*), but not perfectly, as it is possible for an RNA spot to be simultaneously close to the periphery and cell centroid with certain cell shapes

such as a 'dumbbell.' Only the ranking step is different between the peripheral and central metrics; all downstream steps are identical.

Under the null hypothesis that a gene is not subcellularly patterned within a cell, the peripheral and central gene/cell scores have an expected value of 0 and a calculable variance that depends on the number of RNA spots. These statistical underpinnings of the gene/cell scores allow for the identification of spatially significant patterning within gene/cell-types (Methods). Under the null which is each spot's gene identity is drawn uniformly from the set of gene/spots observed from the cell, gene/**cell** scores for k cells of a single cell-type and gene g are independent random variables $X_{g,1}, X_{g,2}, X_{g,3}, ..., X_{g,k}$ with expected values, $\mu_i = 0$ and variance $\mu_i$. Independence in this case comes from the assumption that the scores of a given gene across different cells do not influence each other. Note that the scores of different genes within the same cell are not independent due to the ranking procedure. We define $Y = mean(X_{g,1}, X_{g,2}, X_{g,3}, ..., X_{g,k})$ as the SPRAWL gene/**cell-type** score and a z-score can be calculated under the null that within a cell, each spot's gene identity is exchangeable with the Lyapunov Central Limit Theorem (**Billingsley, 1995**) (Methods: SPRAWL gene/cell-type scoring).

The resulting values y are used to calculate two-sided p-values using the CDF of the standard normal. Multiple hypothesis testing from the numerous gene/cell-type pairs is controlled using the Benjamini-Hochberg correction (**Benjamini and Hochberg, 1995**).

## SPRAWL peripheral and central metric definition

Each gene-cell pair is assigned a SPRAWL score by (1) ranking all RNA spots, (2) calculating median ranks per gene, and (3) normalizing by the expected median rank. Consider a single cell, with a single z-slice, that has $n$ total RNA spots, and $g$ unique genes with each gene having $m_1, m_2, m_3, ..., m_g$ spots such that $\sum_i^g m_i = n$.

For the peripheral metric, let $d_1, d_2, d_3, ..., d_n$ represent the minimum euclidean distance to the periphery of each RNA spot, for the central metric these distances are instead measured from the cell centroid. Each spot is assigned a rank from 1 to n such that the spot with rank 1 is $argmin\left(d|1, d_2, d_3, ..., d_n\right)$ and the spot with rank $n$ is $argmax\left(d|1, d_2, d_3, ..., d_n\right)$ randomly breaking ties where needed.

The ranks are then grouped by gene to calculate the median ranks $t_1, t_2, t_3, ..., t_g$. The peripheral/central SPRAWL gene/cell score $x_i$ for $1 \leq i \leq g$, is the median rank $t_i$ normalized by the expected median rank $t_e$ which is $(n+1)/2$ for all genes independent of $m_i$:

$$x_i = \frac{t_e - t_i}{t_e - 1}$$

Note that $-1 \leq x_i \leq 1$ since $min\left(t|i\right) = 1$ yields $x_i = 1$, and $max\left(t|i\right) = n$ yields $x_i = -1$.

To generalize the definition of the peripheral/central SPRAWL score in the case that a cell has multiple z-slices with a unique cell-boundary and set of spots for each, the distances $d_i$ are calculated from each RNA spot to the cell-boundary/centroid in the same z-slice, and then the ranks are assigned across all z-slices.

## SPRAWL radial metric definition

The radial SPRAWL score is assigned to each gene-cell pair by performing gene-label swapping boot-strapping iterations and measures the tendency of genes to be in one sector of a cell or to be radially dispersed.

Consider a single cell, with a single z-slice, that has $n$ total RNA spots, and $g$ unique genes with each gene having $m_1, m_2, m_3, ..., m_g$ spots such that $\sum_i^g m_i = n$. We restrict to $m_i > 2$ since genes with a single spot do not conceptually have a radial bias.

Before permuting the gene labels, we randomly select a pair of RNA spots for each gene and measure the angle between them with respect to the cell-boundary centroid. Let $\theta$ represent the minimum angle formed by the three points of the location of RNA spot 1 $\left(x_1, y_1\right)$, the cell centroid $\left(x_c, y_c\right)$, and RNA spot 2 $\left(x_2, y_2\right)$. The cell centroid $\left(x_c, y_c\right)$ is approximated as the mean of all vertices in the cell boundary polygon. This process is repeated 10 times and averaged to calculate the mean observed angle of each gene.

The same process is repeated after randomly swapping gene labels but keeping the RNA spot locations the same. We perform 1000 bootstrap iterations. These mean permuted angles serve as the null distribution of mean angles which are used in conjunction with the mean observed angle to

calculate both mean and variance. In the case that a cell has multiple z-slices, the mean cell centroid over all slices is used to calculate pairwise angles without regard to z-slices.

## SPRAWL punctate score definition

The punctate SPRAWL score is conceptually identical to the radial score and also relies on bootstrapping. The punctate score is assigned to each gene-cell pair measuring euclidean distances instead of angles between randomly selected gene pairs. The null distribution is created using the same process as the radial score. In the case that a cell has multiple z-slices, the scoring is performed by projecting all points onto the same (x,y) plane before measuring euclidean distances. This simplification can be readily replaced with true 3D pairwise distances.

## Theoretical features of the SPRAWL peripheral score

While the punctate and radial metrics are calculated using bootstrapping and estimated statistics, the SPRAWL peripheral and central metrics have known properties under the null hypothesis that the gene of interest is not spatially regulated in the given cell. Under this null, the ranks of the gene of interest are chosen with equal probability. In an alternate hypothesis such as a gene being peripherally localized in a cell, RNA spots of the gene of interest will have a skewed probability of being assigned lower ranks, closer to the cell boundary. Under the null hypothesis $E[X_i] = 0$, since $E[T_i] = t_e$ for gene $1 \leq i \leq g$.

$Var[X_i]$ depends on the total number of RNA spots in a cell $n$, and the number of spots of the gene $i$, $m_i$. For example, in the extreme case where $m_i = n$, every spot in a cell is the gene of interest, and $Var[X_i] = 0$. $Var[X_i]$ for any gene can be calculated under the null by iterating over all possible values of $x \in X$ since $X$ is a discrete R.V.

$$Var[X] = \sum_x P\left(T = t_e + x(1 - t_e)\right) \cup t_e - x(1 - t_e)$$

When there are an odd number of gene spots $m$, $t$ is the $(m+1)/2$ rank order statistic, and under the null hypothesis where the ranks are chosen uniformly, the probability of the $r$-th order statistic taking the value $t$ equals:

$$P(T = t) = \frac{\dfrac{t-1}{r-1} \dfrac{n-t}{m-r}}{\dfrac{n}{m}}$$

Where $n$ is the total number of RNA spots, $m$ is the number of RNA spots for the gene of interest, and $r = (m+1)/2$.

$$P(R_0 = r) = \frac{\dfrac{r-1}{\frac{m+1}{2} - 1} \dfrac{n-r}{m - \frac{m+1}{2}}}{\dfrac{n}{m}}$$

Calculating $Var[X]$ when $m$ is even-valued requires significantly more calculation. We still need to calculate

$$P\left(X^2 = x^2\right) = P$$

But $t$ is no longer an order statistic and does not have a closed-form calculation. Instead, $t$ is the average of the 'left of center' $\left(\frac{m}{2}\right)$-th order-statistic $L$, and the 'right of center' $\left(\frac{m}{2} + 1\right)$-th order statistic $R$. Then for a given $x$ and corresponding $t$:

$$P(T = t) = P\left(\frac{L + R}{2} = t\right)$$

We calculate $P(T = t)$ by summing the probabilities of observing all possible pairs of $L$ and $R$ that sum to $2t$. We can think of starting $L$ and $R$ as close to $t$ as possible, and then 'walking' $L$ and $R$ away

from $t$ one rank at a time in lockstep summing over all $i$'s such that $1 \leq t - 1 \leq n$ where $n$ is the total number of spots in the cell:

$$P\left(\frac{L+R}{2} = t\right) = \sum_i P\left(L = t - i \cap R = t + i\right) = \sum_i P\left(R = t + 1 \vee L = t - 1\right) P\left(L = t - 1\right)$$

Omitted for clarity, the ceiling of $t - 1$ is taken and the floor of $t + 1$ above to account for non-integer $t$.

$P\left(R = r \vee L = l\right)$ has an intuitive interpretation that simplifies to an order statistic probability. Since we observe $L = l$ we know that the $R$-th order statistic must be one of the ranks between $l + 1$ and $n$ inclusively. We can renumber these ranks to be between $1$ and $n - l + 1$ and we are interested in the probability that the $1$-st order statistic takes the value $r - l$ in the renumbering. This has the same closed-form solution as described in the odd-valued $m$ case.

Computing $Var\left[X\right]$ for even-valued $m$ is $O\left(n^2\right)$ since we have to iterate over all possible medians, and then for each median we have to 'walk' $L$ and $R$ outwards which is itself $O\left(n\right)$. In comparison, the computation of $Var\left[X\right]$ for odd-valued $m$ is $O\left(n\right)$. Through various optimizations, multiprocessing, and caching, SPRAWL calculated $Var\left[X\right]$ in under an hour for all processed samples.

## SPRAWL is not highly sensitive to exact cell boundary segmentation

Sensitivity of SPRAWL to segmentation and cell-boundary locations was tested by computationally shrinking the cell-boundaries. Median peripheral scores per gene/cell-type were significantly correlated between original cell-boundaries and shrunk cell-boundaries with a Pearson correlation coefficient of 0.85 on the mouse motor cortex datasets (*Figure 1—figure supplement 1e*), suggesting empirically that SPRAWL would have low sensitivity to potential cell-segmentation errors.

## SPRAWL gene/cell-type scoring

Consider a cell-type with k cells with non-zero counts of a gene of interest where each cell is assigned a SPRAWL score $X_1, X_2, ..., X_k$. Note that the $X_i$ are not i.i.d. due to having different $Var\left[X_i\right]$ resulting from different values of $m$ and $n$ as described above.

However, we do make the assumption that the $X_i$ are independent, which has the biological interpretation that the localization of the gene of interest in one cell does not depend on its localization in another cell. Under this assumption, we utilize the Lyapunov Central Limit Theorem *Billingsley, 1995* to estimate that

$$\lim_{k \to \infty} \frac{1}{\sqrt{\sum_{i=1}^{k} \sigma_i^2}} \sum_{i=1}^{k} \left(X_i - \mu_i\right) \to N\left(0, 1\right) \text{ in distribution.}$$

Under the assumption of bounded variance of the $X_i$ satisfying Theorem 27.2 and Corollary 27.3 from *Billingsley, 1995*:

$$\lim_{k \to \infty} \frac{1}{\left(\sum_{i=1}^{k} \sigma_i\right)^{2+\delta}} \sum_{i=1}^{k} E\left[\left|X_i - \mu_i\right|^{2+\delta}\right] = 0$$

We approximate values of $y$ for each gene/cell-type using the observed $x_i$ and theoretical mean and variance whose calculation is described above. These $y$ are used to calculate two-sided p-values from the CDF of the standard normal.

Multiple hypothesis testing over all gene/cell-type pairs is controlled using the Benjamini-Hochberg correction (*Benjamini and Hochberg, 1995*) at a significance level of $\alpha = 0.05$.

We calculate the effect size for each gene/cell-type as the mean gene/cell score $\frac{1}{k} \sum_i^k x_i$.

## SPRAWL is highly specific in identifying genes with subcellular patterning conditional on cell boundaries

If the gene labels of RNA spots within cells of real datasets are permuted to remove any underlying spatial patterning (Methods), none of the metrics detect significant gene/cell-type patterning after Benjamini Hochberg (BH) multiple hypothesis correction with an FDR of 0.05 for any of the four

datasets tested (*Benjamini and Hochberg, 1995*). All metrics were observed to produce uniform p-values under this null dataset regardless of the number of cells per cell-type, as indicated by theory. The median score per gene/cell-type is dependent on the number of cells, with larger groups having median scores closer to zero (*Figure 1—figure supplement 1*). The lack of any false positive calls under the permuted null is consistent at an FDR of 0.05.

## SPRAWL filtering

For all datasets sparse cells and cell-types were filtered out by removing cells with fewer than 10 unique genes and/or fewer than 200 unique RNA spots. Gene/cell-type pairs with fewer than 20 cells were removed from consideration. Further filtering for the radial and punctate metrics requires the removal of genes from cells that have only a single RNA spot. These spots are removed and then the remaining spots can still be scored in this cell for other genes. All filtering steps are implemented as user-accessible parameters and have made SPRAWL more conservative, increasing the confidence of positive hits, but reducing the power to detect real localization differences that occur for lowly expressed genes and/or rare cell-types.

## ReadZS usage and modifications

The ReadZS (*Meyer et al., 2021*) detects read buildup differences between cell-types from single-cell RNA-seq datasets in an annotation-independent manner using equal-sized windows tiling the genome. We modified the ReadZS to analyze at the 3' UTR-level of just the ~250 genes imaged in the BICCN MOp dataset. The 10 X scRNA-seq data was processed individually for the four different mouse donors while the SS2 cells across 45 donors were processed as a single sample due to limited cell counts per mouse.

## Correlation analysis between SPRAWL and ReadZS for MERFISH MOp datasets

For a given SPRAWL gene and spatial metric, the median ReadZS score of that gene for each cell-type was correlated against the median SPRAWL score over the same cell-types. For positive-strand genes, a larger ReadZS score indicates longer 3' UTR isoforms, and vice versa for negative-strand genes. A proxy for 3' UTR length was defined as the distance between the annotated start of the 3' UTR and the RNA mapping position. The span in estimated 3' UTR lengths was measured as the difference between the longest and shortest median cell-type 3' UTR proxy lengths.

## Vizgen Brainmap and Liver showcase clustering to produce cell-type proxies

Neither the Vizgen MERFISH Mouse Brain Map nor Liver showcase datasets provided cell-type annotations. We decided to roughly cluster the cells into groups to serve as a proxy for cell-type. The Leiden clustering method was used to find well-connected clusters in all of the filtered 90% highest spot-count cells using the Scanpy python package (*Wolf et al., 2018*). First, each dataset was normalized so that each cell had 10,000 spots, then the top 40 principal components were used to build the neighborhood graph with 10 neighbors and perform the Leiden clustering. This resulted in 22 clusters for the Brainmap dataset and 100 clusters for the Liver dataset. The fraction of cells in each cluster was consistent across biological replicates for the Vizgen Liver (*Figure 3—figure supplement 1*) and Vizgen Brainmap (data not shown) indicating that cells were primarily clustering by type, and not by batch. To estimate the batch effect, we calculated the probability that two cells originated from the same biological replicate given that they were in the same cluster, and compared this to the overall probability that two cells are from the same biological replicate. All clusters were within 0.05 of the overall probability of two cells sharing a batch.

## Simulations to benchmark SPRAWL sensitivity and specificity

Null simulated datasets were created from the MERFISH BICCN spatial dataset by randomly permuting the RNA-spot gene labels within each cell across the entire dataset. The cell-boundaries, RNA-spot counts, and RNA (x,y,z) coordinates were preserved in the null dataset.

## Identification of RBP and miRNA binding to Timp3 3' UTR

The RNAInter v4.0 RNA interactome repository was used to search for RBPs and miRNAs with experimental evidence of binding in the 3' UTR of the *Mus musculus Timp3*, *Slc32a1*, *Cxcl14*, and *Nxph1* genes (*Kang et al., 2022*). Target regions for RBPs were taken from RNAInter, while miRNA binding sites were generated and cross-checked against TargetScan release 8.0 (*McGeary et al., 2019*) and miRWalk (*Sticht et al., 2018*). Only miRNAs shared by RNAInter, TargetScan, and miRWalk results with experimental evidence were considered.

## RNAs with signal peptides do not have significant central or peripheral localization

We hypothesized that RNAs encoding a signal recognition peptide (SRP) for translation on the rough endoplasmic reticulum would be nuclear localized and would, therefore, be more centrally localized than genes without signal peptides. We predicted the presence of SRPs using DeepSig (*Savojardo et al., 2018*) with protein sequences downloaded from Gencode release M28 protein-coding transcripts fasta for all genes present across the MOp, Vizgen Brainmap, and SeqFISH + cortex datasets. For genes with multiple protein isoforms, the longest isoform was selected for SRP prediction. In all datasets, the per-gene per-cell peripheral and central scores were not significantly different according to a Kolmogorov Smirnov test (*Figure 5—figure supplement 1a*).

## Genes enriched in single-nucleus RNAseq are marginally correlated with periphery score

We tested whether nuclear-localizing genes would be assigned higher SPRAWL central periphery scores utilizing both the 10 X single-cell RNAseq (scRNA-seq) as well as 10 X single-nucleus RNAseq (snRNA-seq) from the BICCN consortium (*BRAIN Initiative Cell Census Network (BICCN), 2021*). The single-cell sequencing data was first normalized to the number of counts per gene per cell per one million (TPM) reads for both the cell and nuclear datasets. The median gene/cell-type TPM for both sequencing datasets was determined, and the nuclear-fraction score was determined to be snRNA-seq-TPM/(snRNA-seq-TPM +scRNA-seq TPM). The median periphery score per gene/cell-type was correlated against the median snRNA-seq-TPM, scRNA-seq-TPM, and nuclear-fraction. In all comparisons, the correlation coefficients were small in magnitude, but were significantly positive for the snRNA-seq, indicating a link between X tendency and peripherality, and significantly negative in the nuclear-fraction analyses, indicating a link between the gene's enrichment in nuclear reads and its distance from the cell periphery. The small effect size was detectable due to the approximately 8000 gene/cell-type data points and provides weak support for the hypothesis. We investigated which genes, if any, are differentially nuclear-enriched across cell-types by sequencing and concordantly by peripheral score and discovered *Wipf3* (*Figure 5—figure supplement 1b*) and *Slc30a3*, which were highly negatively correlated with mean Pearson correlation coefficients of –0.86 and –0.93 across MERFISH MOp samples. Surprisingly, *Satb2* was also discovered to be significant, but had a highly positive mean Pearson correlation coefficient of 0.95. All genes were determined to be significant after Benjamini Hochberg's multiple hypothesis correction.

## Pericyte culture experimental setup with ELISA, qPCR, and BCA readouts

Human brain vascular pericytes (PCs, Sciencell) were cultured up to passage 5 in low-glucose DMEM (Gibco) supplemented with 10% FBS. ~$1.2 \times 10^5$ PCs were seeded in each well of a six-well plate pre-coated with 0.1% gelatin. PC lysates and conditioned media were collected 6 hr after seeding for RNA isolation and ELISA applications. Similar samples were collected on 24, 48, 72, and 120 hr after seeding. The 120- hr timepoint was not considered for analysis since the cells had lifted off from the culture dish. RNA was isolated with the PureLink RNA Kit (Invitrogen) and reverse transcribed with the iScript cDNA Synthesis Kit (Bio-Rad) and qRT-PCR was performed on a CFX96 Real-Time System (Bio-Rad) using SsoAdvanced Universal supermix (Bio-Rad). Transcript levels of *TIMP3* with short or long

3' UTR relative to housekeeping gene (*B-actin* or *GAPDH* or *18* S rRNA) were determined for each timepoint with four biological replicates and three technical replicates.

ELISA measurements were made using the Human TIMP-3 ELISA Kit from Invitrogen (Catalog # EH458RB) and precisely following the manufacturer's instructions.

>Proximal_primer_1_fwd
GGGAACTATCCTCCTGGCCC
>Proximal_primer_1_rev
TTCTGGCATGGCACCAGAAAT
>Proximal_primer_2_fwd
AGGTCTATGCTGTCATATGGGGT
>Proximal_primer_2_rev
TGGGGCCAGGAGGATAGTTC
>Distal_primer_1_fwd
AATTGGCTCTTTGGAGGCGA
>Distal_primer_1_rev
GCGGATGCTGGGAGAATCTA
>Distal_primer_2_fwd
TAGCCAGTCTGCTGTCCTGA
>Distal_primer_2_rev
GGGTTCGAGATCTCTTGTTGG

## Timp3 protein production estimation

An estimate of the rate of Timp3 protein production per cell per hour was calculated using the ELISA Timp3 measurements and cell counts at each hour. The extracellular Timp3 concentration from the ELISA measurements was converted from ng/mL to ng's of Timp3 per cell using the known culture volume of 2 mLs and the cell counts at the same timepoint. This value represents the amount of extracellular Timp3 per cell; in order to calculate how much Timp3 is produced, the amount of degraded Timp3 between timepoints is estimated from the tissue-culture half-life estimate of 15 hr (*Mao et al., 2021*). The Timp3 protein production per cell at time t2 is estimated to be the difference between the amount of Timp3 at t2 and the previous timepoint t1, plus the degraded Timp3 fraction from t1, divided by the number of cells at t2.

## qPCR analysis of pericyte culture Timp3 3' UTR abundance

Our goal is to estimate the relative abundance of the short vs. long *TIMP3* 3' UTR isoforms at multiple timepoints during cell culture. The ratio of short to long *TIMP3* 3' UTR isoform in a sample can be estimated using the proximal and distal qPCR primer critical threshold (CT) values. Let the amount of template present in the sample which can be amplified by the proximal qPCR primer be represented as $P$. Similarly let the un-amplified amount of template for the distal primer be represented as $D$.

At the critical threshold number of cycles for both the distal $CT_D$ and proximal $CT_P$ qPCR primers, the absorbances will be equal. Assuming that the initial amount of template $P$ and $D$ doubles in each cycle we can create an equation to solve for the ratio of $\frac{P}{D}$

$$P * 2^{CT_P} = D * 2^{CT_D}$$
$$\frac{P}{D} = \frac{2^{CT_D}}{2^{CT_P}} = 2^{CT_D - CT_P}$$

Since the proximal primers can amplify both the short and long isoforms, while the distal primers can only amplify the long isoforms we can rewrite the previous equation with $S$ and $L$ representing the amount of short and long *TIMP3* 3' UTR template in each sample.

$$\frac{S + L}{L} = 2^{CT_D - CT_P}$$

Since $S > 0$ and $L > 0$, we expect $2^{CT_D - CT_P} > 1$, however, we observe 219 of 240 qPCR biological/technical replicates having $2^{CT_D - CT_P} < 1$.

We at first considered that this discrepancy may be due to differences in the amplification efficiency of the proximal and distal qPCR primers which are assumed to be equal and 100% efficient with a doubling in each PCR cycle. However, if for some reason the proximal and distal primers had different efficiencies, it would be incorrect to directly compare their CT values. We estimated the efficiencies of the proximal 1, proximal 2, distal 1, and distal 2 qPCR primers by measuring the CT values at twofold dilutions of the same cDNA template and observed that all primer pairs had near 100% efficiency except for proximal primer 1 which had 82% efficiency (*Figure 6—figure supplement 1*). For the qPCR analyses presented in this paper, proximal primer 2 and distal primer 2 were used. Efficiency calculations were made by finding the slope, m, of the line of best fit for (x=log2 cDNA dilution) vs. (y=CT), and then converting slope to efficiency as (100/2^(m-1)).

Given that qPCR efficiency is not the cause of the widely observed $\frac{S+L}{L} < 1$ ratios, we believe that the existence of a template which is only amplified by the distal and not the proximal qPCR primer pairs could be confounding. Such templates could arise from incomplete reverse transcription or spliced Timp3 3' UTR isoforms. While we do not have a way to control for this in the current qPCR experiment, we might expect to observe the same external effect at each timepoint.

## Acknowledgements

We'd like to acknowledge the Salzman lab members for helpful discussion and suggestions, especially Elisabeth Meyer, Roozbeh Dehghannasiri, and Tavor Baharav for text edits as well as Jonathan Liu from the Chan-Zuckerberg Biohub. We acknowledge George Emmanuel for his help in the initial data download and processing. We acknowledge Pehr Harbury and Mark Krasnow for feedback and Mark Krasnow and Catherine Blish for the human lung ex-situ culture time-course datasets. Some of the computing for this project was performed on the Sherlock cluster. We would like to thank Stanford University and the Stanford Research Computing Center for providing computational resources and support that contributed to these research results. We would like to thank funding sources from the NCI (5F31CA243170-02), the NHGRI (1R56HG011231-01), and NIGMS (1R35GM139517-01). This research was supported in part by a training grant from the NIH Cellular and Molecular Training Grant (NIGMS, grant number 5T32GM007276). Support also came from (R35HL150766, 1R21NS123469 to DMG), American Heart Association (Established Investigator Award, 19EIA34660321 to DMG and Career Development Award, 856332 to JMD).

## Additional information

### Funding

| Funder | Grant reference number | Author |
|---|---|---|
| National Cancer Institute | 5F31CA243170-02 | Rob Bierman |
| National Institute of General Medical Sciences | 1R35GM139517-01 | Julia Salzman |
| National Heart, Lung, and Blood Institute | R35HL150766 | Daniel M Greif |
| National Institute of General Medical Sciences | 5T32GM007276 | Rob Bierman |
| National Human Genome Research Institute | 1R56HG011231-01 | Julia Salzman |
| American Heart Association | 19EIA34660321 | Daniel M Greif |
| American Heart Association | 856332 | Jui M Dave |
| National Institute of Neurological Disorders and Stroke | 1R21NS123469 | Daniel M Greif |

| Funder | Grant reference number | Author |
|--------|------------------------|--------|

The funders had no role in study design, data collection and interpretation, or the decision to submit the work for publication.

## Author contributions

Rob Bierman, Data curation, Software, Formal analysis, Investigation, Visualization, Methodology, Writing - original draft, Writing - review and editing; Jui M Dave, Formal analysis, Validation, Investigation, Methodology, Writing - original draft; Daniel M Greif, Resources, Supervision, Validation, Methodology, Writing - review and editing; Julia Salzman, Conceptualization, Resources, Formal analysis, Supervision, Funding acquisition, Investigation, Methodology, Project administration, Writing - review and editing

## Author ORCIDs

Rob Bierman ⓘ https://orcid.org/0000-0001-8513-7425
Daniel M Greif ⓘ https://orcid.org/0000-0002-9842-3751
Julia Salzman ⓘ https://orcid.org/0000-0001-7630-3436

Reviewer #1 (Public Review): https://doi.org/10.7554/eLife.87517.2.sa1
Reviewer #2 (Public Review): https://doi.org/10.7554/eLife.87517.2.sa2
Reviewer #3 (Public Review): https://doi.org/10.7554/eLife.87517.2.sa3
Author response https://doi.org/10.7554/eLife.87517.2.sa4

---

# Additional files

## Supplementary files

• Supplementary file 1. Counts of unique, significant, and opposite-effect genes in each experiment/metric combination. Genes are defined as significant if they are observed to be significant in at least cell-type in any replicate. Opposite-effect genes are those observed to have at least one significantly positive Subcellular Patterning Ranked Analysis With Labels (SPRAWL) gene/cell-type score, and one significantly negative SPRAWL gene/cell-type score.

• MDAR checklist

## Data availability

The current manuscript is mostly a computational study, but the Timp3 pericyte experiment did generate data which is available in the SPRAWL GitHub, (copy archived at *Bierman, 2024*).

The following previously published datasets were used:

| Author(s) | Year | Dataset title | Dataset URL | Database and Identifier |
|-----------|------|---------------|-------------|-------------------------|
| Company V | 2022 | Vizgen MERFISH Mouse Brain Map (BrainMap) | https://console.cloud.google.com/marketplace/product/gcp-public-data-vizgen/vizgen-mouse-brain-map | Google Cloud, vizgen-mouse-brain-map |
| Company V | 2022 | Vizgen MERFISH Liver showcase | https://info.vizgen.com/mouse-liver-data?submissionGuid=832a9f61-22d3-44c1-a2cf-838c166d9ac5 | Google Cloud, 832a9f61-22d3-44c1-a2cf-838c166d9ac5 |
| Pisco A | 2022 | MERFISH mouse comparison study | https://figshare.com/projects/MERFISH_mouse_comparison_study/134213 | figshare, 134213 |

*Continued on next page*

*Continued*

| Author(s) | Year | Dataset title | Dataset URL | Database and Identifier |
|---|---|---|---|---|
| CHL Eng, Lawson M, Zhu Q, Dries R, Koulena N, Takei Y, Yun J, Cronin C, Karp C, Yuan GC, Cai L | 2019 | SeqFish+ dataset | https://github.com/CaiGroup/seqFISH-PLUS/blob/master/sourcedata.zip?raw=true | GitHub, ee6c416 |
| Zhang M, Eichhorn SW, Zingg B, Yao Z, Cotter K, Zeng H, Dong H, Zhuang X | 2021 | MERFISH primary mouse cortex (MOp) dataset | https://download.brainimagelibrary.org/cf/1c/cf1c1a431ef8d021/processed_data/ | Brain Image Library, cf1c1a431ef8d021 |

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
