## [Editor Report · eLife assessment]

This paper describes an **important**, well-organized study into an under-exploited area of spatial transcriptomics. The limitations of the approach are generally made clear, but there is insufficient orthogonal validation to demonstrate the biological significance of the results, which leads to the evidence for the claims being currently **incomplete**. Nevertheless, the tools presented will provide a resource to researchers wishing to characterise spatial patterning of mRNAs, and the paper will be of interest to researchers studying cell biology, RNA biology, and method development for spatial transcriptomics/proteomics.

---

## [Referee Report · Reviewer #1 (Public Review)]

Bierman et al. have developed a set of metrics for measuring the spatial patterning of mRNAs in high-throughput fluorescence in situ hybridisation experiments and applied these to identify a subset of mRNAs whose spatial patterning correlates with 3'UTR length. A strength of the study is the clarity and honesty with which the authors have outlined the strengths and weaknesses of their own approach and reported negative results. A key benefit of the tool is that the methodological choices allow wide applicability to existing datasets. However, these choices also feed into a limitation of the method, which is the difficulty in interpreting the biology underpinning the metrics - raising the question of how users will understand the output of the tool.

---

## [Referee Report · Reviewer #2 (Public Review)]

The authors develop SPRAWL (Subcellular Patterning Ranked Analysis With Labels), a statistical framework to identify cell-type specific subcellular RNA localization from multiplexed imaging datasets. The tool is able to assign to each gene and in each annotated cell type, a score (with a p-value) that measures:

- Peripheral/central localization of RNAs within the cell, based on a previous segmentation step defining cell boundaries and the centroid coordinate.

- Radial/punctuate localization of RNAs within the cell

The method is applied to three multiplexed imaging datasets, identifying defined and cell-type specific patterns for several transcripts.

In the second part of the manuscript, the authors couple SPRAWL with ReadZS, a computational tool developed by the same group and recently published (Meyer et al, 2022). Starting from single-cell datasets, ReadZS is able to quantify 3'UTR length in each cell type. The authors find a subset of genes showing a positive, or negative correlation between the predicted localization and the predicted 3'UTR length across cell types.

Strengths:

As the authors state in the introduction, the study of subcellular RNA localization, with the characterization of organizational principles and of molecular regulation mechanisms, is extremely relevant. The authors develop a strategy to detect statistically significant and non-random patterns of RNA sub-cellular localization in MERFISH and SeqFISH+ datasets, i.e. emerging platforms producing spatially resolved maps of hundreds of transcripts with cellular resolution.

Weaknesses:

Although the method and the presented results have strengths in principle, the main weakness of the paper is that these strengths are not directly demonstrated. That is, insufficient validations are performed to show the biological significance of the results and to fully support the key claims in the manuscript by the data presented.

In particular, the authors imply that their tool is unique and not comparable to any other method. Therefore there is no comparison of SPRAWL with any other method. For example, a comparison could be made with Baysor (Petukhov, V et al. Nat Biotechnol. https://doi.org/10.1038/s41587-021-01044-w). According to the authors, this method is able to identify "small molecular neighbourhoods with stereotypical transcriptional composition" and provides a "General approach for statistical labeling of spatial data".

The authors claim that SPRAWL is able to identify spatial patterns of localization and generated relevant hypotheses to be tested, yet the manuscript contains little proof that the results have biological significance (for example association of RNAs with specific subcellular compartments) and there is no experimental validation for the results obtained applying this method.

The correlation between localization scores and 3'UTR length across cell types for certain genes is also not experimentally validated: results are based on inference from single-cell or imaging data, with no complementary experimental validation.

It is therefore very difficult to assess the biological relevance of the results produced by SPRAWL.

---

## [Referee Report · Reviewer #3 (Public Review)]

Bierman et al. present a novel statistical framework for examining the subcellular localisation of RNA molecules. Subcellular Patterning Ranked Analysis With Labels, SPRAWL, uses the data available in multiplexed single-cell imaging datasets to assign four metrics of localisation patterns to RNA at a gene per cell level. These easy-to-understand scores, ranging from -1 to 1, can be averaged to detect cell-type specific spatial patterns or used in tandem with tools for RNA 3' UTR length or splicing state to determine the correlation between subcellular localisation and RNA isoforms. Such quantitive association between RNA isoforms and localisation provides a useful tool to determine candidate genes for future studies.

The peripheral and central scores indicate the proximity of RNA molecules to the cell boundary and centre of the cell respectively in relation to other RNA present in the cell. Whilst understanding whether a gene tends to be localised to the cellular membrane is important, it is unclear what biological benefits the central metric gives compared to high "anti-peripheral" scores considering that no single organelle (eg. the nucleus) is located specifically at the centre of the cell in all cell-types.

The punctate and radial patterning scores provide information on the spatial aggregation of RNA molecules of a given gene within a cell. Whilst the punctate score is easy to understand as simply the distance between RNA, the radial score, the angle between RNA, is harder to understand from the main text and would benefit from a schematic showing how this is in respect to the cell-boundary centroid.

Despite endeavouring to create a robust statistical measure of RNA subcellular localisation, this paper is full of inconsistencies. Values (eg. Pearson correlation coefficient values, number of significant genes, number of total genes) and names (eg. cell types, gene names) stated throughout the main text and figures/table do not match repeatedly and without fixing these disparities, the conclusions from this paper are hard to believe.

---

## [Author Response]

**Reviewer #1:**

We agree with Reviewer 1 that the flexibility of SPRAWL also makes it difficult to interpret its outputs. We consider SPRAWL to be a hypothesis-generation tool to answer simple questions of subcellular localization in a statistically robust manner. In this paper we include examples of how it can be incorporated with other tools and wetlab experimentation to build biological intuition. Our hope is that the SPRAWL software, or even the underlying simple statistical ideas are of use to others in the field.

**Reviewer #2:**

We agree with Reviewer #2 that this manuscript does not demonstrate biological significance of the observed results of applying SPRAWL to massively multiplexed FISH datasets. We agree it would require additional wetlab experiments such as cell-type specific and isoform-resolved fluorescence in-situ hybridization, which we consider beyond the scope of this paper. We believe that the observed correlations of subcellular localization detected by SPRAWL and the differential 3’ UTR usage detected by ReadZS are compelling, although not conclusive, as are the Timp3 experimental studies.

Our understanding is that Baysor is primarily a cell-segmentation algorithm, which is not what SPRAWL attempts to achieve. Baysor states that it identifies “cells of a distinct type will give rise to small molecular neighborhoods with stereotypical transcriptional composition, making it possible to interpret such neighborhoods without performing explicit cell segmentation” which we understand to mean that Baysor identifies spatial groupings of cells with “stereotypical transcriptional composition” rather than subcellular RNA localization. We do not think that SPRAWL and Baysor are comparable, but instead Baysor could be used as an upstream step to SPRAWL to potentially improve cell segmentation.

**Reviewer #3:**

We thank Reviewer #3 for identifying discrepancies in the paper which we addressed to the best of our abilities.